# Invasion of vaginal epithelial cells by uropathogenic *Escherichia coli*

John R. Brannon [1✉], Taryn L. Dunigan[1], Connor J. Beebout [1], Tamia Ross[1], Michelle A. Wiebe[1], William S. Reynolds[2] & Maria Hadjifrangiskou [1,2,3✉]

Host-associated reservoirs account for the majority of recurrent and oftentimes recalcitrant infections. Previous studies established that uropathogenic *E. coli* – the primary cause of urinary tract infections (UTIs) – can adhere to vaginal epithelial cells preceding UTI. Here, we demonstrate that diverse urinary *E. coli* isolates not only adhere to, but also invade vaginal cells. Intracellular colonization of the vaginal epithelium is detected in acute and chronic murine UTI models indicating the ability of *E. coli* to reside in the vagina following UTI. Conversely, in a vaginal colonization model, *E. coli* are detected inside vaginal cells and the urinary tract, indicating that vaginal colonization can seed the bladder. More critically, bacteria are identified inside vaginal cells from clinical samples from women with a history of recurrent UTI. These findings suggest that *E. coli* can establish a vaginal intracellular reservoir, where it may reside safely from extracellular stressors prior to causing an ascending infection.

[1] Department of Pathology, Microbiology and Immunology, Division of Molecular Pathogenesis, Nashville, TN, USA. [2] Department of Urology, Nashville, TN, USA. [3] Vanderbilt Institute for Infection, Immunology & Inflammation, Vanderbilt University Medical Center, Nashville, TN, USA. ✉email: john.r.brannon@vumc.org; maria.hadjifrangiskou@vumc.org

The presence of bacterial reservoirs, particularly those within the host, are largely responsible for the recurrence of many infections[1–7]. Urinary tract infections (UTIs), the most prevalent bacterial infection in human adults, are characterized by high levels of recurrence, with ~40% of infected individuals experiencing a recurrent UTI (rUTI) within 6 months of the original episode[8,9]. Moreover, a range of 50–80% of rUTI patients are infected repeatedly by the same bacterial strain, indicating the presence of reservoirs within the host[10–12]. Disease outcomes are diverse, ranging from asymptomatic bacteriuria (ASB) to symptomatic infection of the bladder (cystitis), and kidneys (pyelonephritis), both of which can progress to bacteremia, or sepsis in vulnerable populations[9,13]. Treatment of rUTIs drives the prescription of antibiotics, making UTIs the second leading reason for most antibiotic prescriptions[13–15]. In this post-antibiotic era, it is pertinent to understand how pathogenic reservoirs contribute to recurrent infection, such as what is observed in rUTIs[16].

Over 80% of UTIs are caused by uropathogenic *Escherichia coli*[13]. To date, three reservoirs are proposed to facilitate rUTI by UPEC: the gut[2,3,17], quiescent intracellular reservoirs (QIRs) within the urothelium[1,18], and the vagina, where *E. coli* can co-exist alongside commensals and adhere to vaginal epithelial cells (VECs)[19–24]. Of these three reservoir niches, the vagina is the least well-characterized as an UPEC transient colonization niche. Extensive studies have elucidated a transient intracellular stage in the bladder infection program of UPEC. Specifically, UPEC can use type 1 pili to adhere to the urothelium[25–28]. Adherence triggers a series of events that lead to UPEC internalization into non-degradative vacuoles[29]. UPEC escape these vacuoles and proliferate into biofilm-like, intracellular bacterial communities (IBCs) within the cytosol[29]. Dispersal from the IBCs in conjunction with the exfoliation of superficial umbrella cells in response to acute infection, exposes the underlying urothelial layers to UPEC, which can then establish what have been termed QIRs[1,29]. QIRs form within undifferentiated transitional bladder epithelial cells and comprise bacteria that are presumed to be metabolically dormant[1]. The QIR allows UPEC to reside within the bladder for extended periods of time, evading host defenses and antibiotic therapy. Accordingly, QIRs are implicated as a potential reservoir for rUTIs.

Considering that UPEC can invade the urothelium and bind to vaginal cells, we sought to better understand host–pathogen interactions at the vaginal interface. Here, we demonstrate that UPEC clinical isolates not only adhere to VECs, but also invade and establish vaginal intracellular communities (VICs). Using tissue culture monolayer- and murine models of acute and chronic UTI, as well as a murine model of vaginal colonization, we found that UPEC invades VECs through a zipper-like mechanism by harnessing a set of host factors distinct from those reported for bladder cell invasion. In human-derived vaginal samples, we identified the presence of UPEC inside VECs from women with a history of rUTI. We propose that VEC invasion and establishment of VICs is a previously unrecognized stage in UTI pathogenesis.

## Results

**UPEC invades VECs**. While extensive studies have elucidated the bladder infection program of UPEC[25–30], the means through which UPEC colonizes other niches remains to be fully revealed. Multiple studies have reported that UPEC colonizes the vagina prior to acute UTI[9,19,31–34]. However, to date, there is limited knowledge into UPEC–vaginal epithelium interactions. We thus investigated the interaction of UPEC with immortalized VK2 E6/E7 VECs, which are often used to model the VEC interactions

with vaginal pathogens and commensal microorganisms[35–39]. We first measured the ability of a prototypical UPEC cystitis strain, UTI89, to associate with cultured VECs. Consistent with previous studies, we observed that UPEC adheres to immortalized VECs at various multiplicities of infection (MOI) and remains stable over time (Fig. 1a, b)[22]. Strikingly, we noted a subpopulation of UPEC that survived a 2-h incubation with the antibiotic gentamicin (Fig. 1a, b). Gentamicin is bactericidal, but cannot permeate eukaryotic membranes, suggesting that the gentamicin-resistant subpopulation is intracellular[25]. Fluorescence microscopy of VK2 E6/E7 cells infected with fluorescently labeled UPEC (UTI89/pCom-GFP) revealed intracellular bacteria that appeared enclosed in vacuoles (Fig. 1c, white arrowheads). This observation was confirmed via transmission electron microscopy (TEM) (Fig. 1d). To further confirm that UTI89/pCom-GFP was truly intracellular, we probed our samples using an *E. coli*-specific antibody (Supplementary Fig. 1a) without prior permeabilization of the vaginal cell monolayer (Fig. 1e). The antibody only stained extracellular UTI89/pCom-GFP and did not stain intracellular UTI89/pCom-GFP (Fig. 1e, white arrowheads).

*E. coli* is a genetically diverse species that spans at least six well-characterized different phylogenetic groups and nine different pathotypes[40,41]. Amongst these, UPEC is known to have the highest level of genetic and phenotypic diversity[40–44]. Considering the level of heterogeneity among UPEC isolates, we next sought to determine whether invasion of the vaginal epithelium is strain-specific. For these studies, a panel of UPEC gentamicin-sensitive clinical strains isolated from female patients with a UTI at Vanderbilt University Medical Center was used[42]. These strains exhibited similar adherence levels, and all exhibited VEC invasion to different levels (Fig. 1f). These data indicate that VEC invasion is not strain-specific, but rather an ability that is broadly shared by urinary *E. coli* isolates.

**E. coli invades vaginal and bladder cells through distinct mechanisms**. Adherence and invasion within the bladder is mediated primarily by type 1 pili, which are adhesive appendages that bind to mannosylated urothelial proteins[25,45]. Urothelial invasion is initiated by binding of the pilus tip adhesin, FimH, to β1 and α3 integrins found on bladder epithelial cells[46]. Previous studies have implicated type 1 pili-mediated adherence to the vaginal epithelium[21,22]. To begin addressing the means by which UPEC adhere to and invade VECs, we first compared adherence and invasion levels in bladder- and vaginal epithelial cell lines. We found no significant difference between the adherence and invasion of UTI89 between the 5637 bladder and VK2 E6/E7 vaginal cell lines (Fig. 2a). In agreement with the previous work[21,22], a *fim* operon deletion mutant UTI89 Δ*fimA-H* (Fig. 2b), displayed significantly reduced adherent titers onto VECs compared to the parental UTI89 strain (Fig. 2b). Notably, adherence and invasion are not completely ablated in the Δ*fimA-H* strain (Fig. 2b), suggesting that in addition to type 1 pili, other adhesive appendages or mechanisms contribute to vaginal colonization and invasion.

Bacterial-induced phagocytosis into nonphagocytic cells such as VECs occurs via a trigger or zipper mechanism, though other forms of endocytosis also exist[47]. In the trigger mechanism, bacteria actively deploy either a type III—typically missing or incomplete in most UPEC strains—or type IV secretion system to translocate effector proteins across the host cell membrane thereby manipulating the host cytoskeleton into drastic, all-or-nothing phagocytosis[47]. By contrast, through the zipper mechanism bacteria adhere to the host cells and passive receptor contact initiates a subtle, graded membrane extension that gradually results in phagocytosis[47]. Both zipper and trigger mechanisms

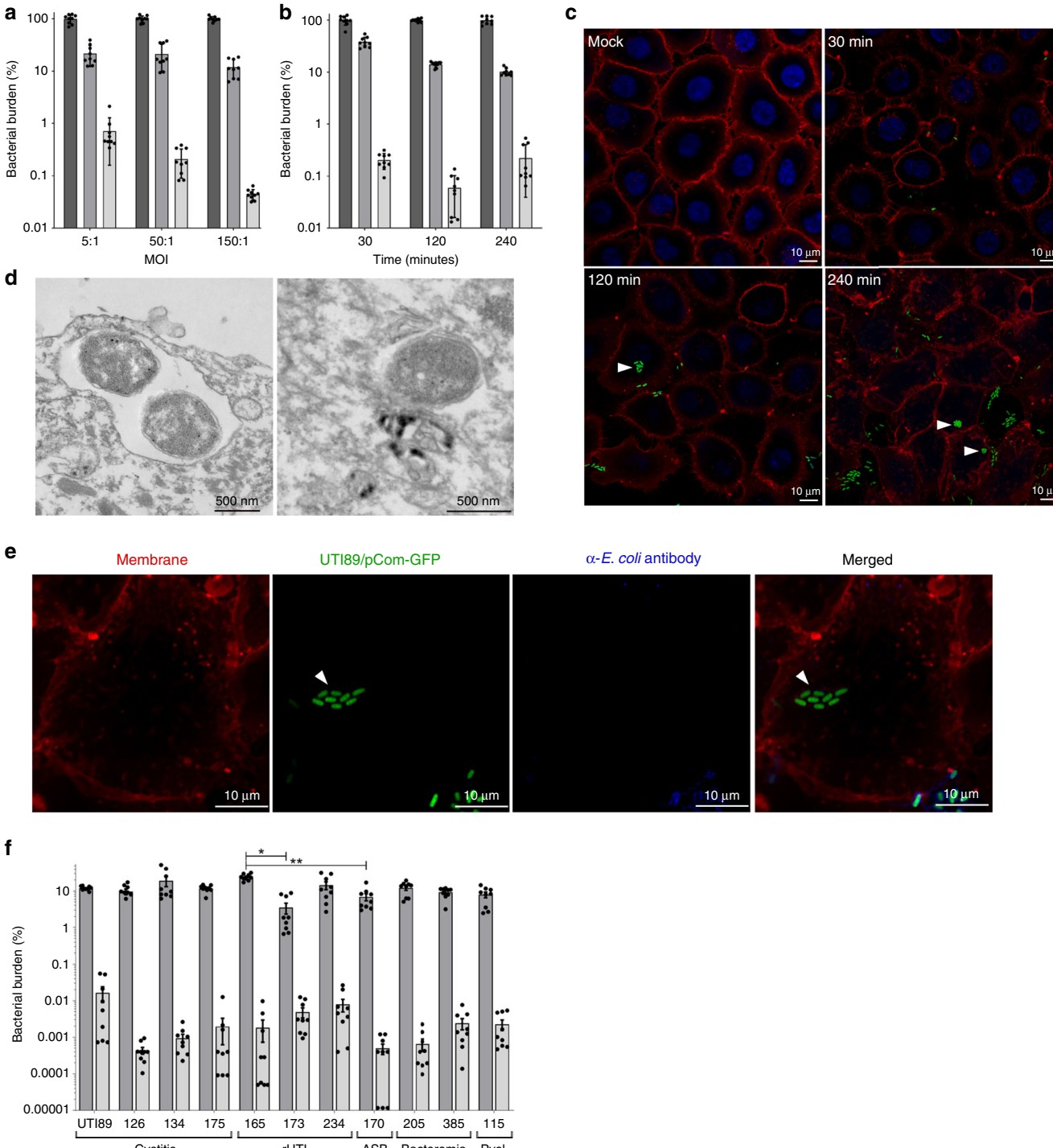

induce actin re-arrangement in the host cell[47]. Staining of infected VK2 E6/E7 cells with phalloidin, an F-actin-specific stain revealed that UPEC adherence resulted in remarkable actin bundling, specifically at the surface near the bacteria (Fig. 2c). Additionally, bacteria within the VEC were surrounded by actin (Fig. 2c). Incubation with 0.1 and 1.0 µg/mL cytochalasin D, which specifically inhibits F-actin filament elongation, significantly reduced invasion of UPEC into VK2 E6/E7 cells (Fig. 2d), without affecting adherence (Supplementary Fig. 1b)[48]. The reduction in invasion by cytochalasin D was reversed by washing VK2 E6/E7 cells prior to the addition of bacteria to the culture (Fig. 2d). These results support that F-actin polymerization is required for UPEC invasion of VECs. Fluorescently labeled latex

beads were only observed on the surface of VECs (Fig. 2c), indicating that the observed invasion is bacteria-specific and non-passive. Paraformaldehyde (PFA)-killed bacteria were also found within VECs (Fig. 2c), suggesting that surface receptors on intact bacterial cells—whether live or dead—appear to mediate invasion into VECs and do not require an active secretion system.

Through the zipper mechanism, UPEC and other bacteria manipulate the host signaling cascades to induce cytoskeletal re-arrangement that leads to bacterial internalization[25,49–51]. Invasion of bladder epithelial cells is known to involve host actin and microtubule re-arrangement through manipulation of the tyrosine kinase, phosphoinositide 3-kinase (PI3K), and histone deacetylases[25,45,49]. To determine whether VEC invasion occurs

**Fig. 1 UPEC adheres to and invades VK2 E6/E7 VECs.** VK2 E6/E7 cells were infected with the indicated strain of UPEC. **a** and **b** Gentamicin-based adherence and invasion assay of VECs infected by: **a** UTI89 at the indicated MOIs for 30 min. **b** UTI89 at a MOI of 5:1 for 30, 120, and 240 min. Planktonic bacteria are washed away by PBS, which leaves behind the adherent bacteria. Gentamicin kills extracellular bacteria; whereas, bacteria that invade cells are safe from the membrane impermeable antibiotic. UTI89 and clinical isolates adhered to and invaded VECs to different extents. Adherent (medium gray bars) and intracellular (light gray bars) bacterial populations are relative to the average of the total population of bacteria (dark gray bars) within the well. **a**, **b** Data for initially setting-up the model are presented as the mean of nine independent experiments with standard deviation to demonstrate the level of consistency. **c** Confocal laser scanning microscopy images of mock infected (PBS) and UTI89/pCom-GFP for 30, 120, and 240 min that were stained with ToPro-3 for DNA and r-WGA to outline VEC membranes. **d** UTI89 at a MOI of 50:1 for 2 h imaged by TEM. TEM images are representative of two independent experiments. **e** Microscopy image of VK2 E6/E7 cells infected with UTI89/pCom-GFP for 120 min were stained with r-WGA; additionally, to further differentiate *E. coli* localization permeabilization was not performed and extracellular *E. coli* cells were stained with α-*E. coli* (blue). White arrows point toward intracellular bacteria. Confocal laser scanning microscopy images are representative of three independent experiments. **f** Low-passage clinical UPEC isolates were utilized in gentamicin-based adherence (medium gray bars) and invasion (light gray bars) assay with VECs for 2 h. **f** Data are the mean of nine independent experiments with error bars representing s.e.m. For statistical analysis two-way ANOVA with Tukey post-hoc test (*P < 0.05, ** P < 0.01).

through the same cytoskeletal actin re-arrangement seen during bladder cell invasion, we first inhibited the activity of PI3K[25]. While the general tyrosine kinase inhibitor, genistein, ablated UTI89 VEC invasion (Fig. 2e), as was reported for bladder epithelial cells, wortmannin, a specific inhibitor of PI3K only marginally impacted VEC invasion (Fig. 2f)[25,52,53]. These results indicate that other host kinases, besides PI3K are essential to VEC invasion. Conversely, addition of the inhibitor PP1, an inhibitor of the Src-family of tyrosine kinases, resulted in a significant reduction in VEC invasion (Fig. 2g); in comparison, PP1 inhibition of the type 1 pili-mediated invasion of bladder cells has been previously reported with mixed results[25,46,54]. To test if the difference in cytoskeletal re-arrangements extended to microtubules or was specific towards actin, we next tested the ability of microtubule inhibitors to disrupt UPEC internalization into VEC. The three drugs: nocodazole, which promotes microtubule depolymerization; vinblastine, another inducer of microtubule depolymerization that also promotes tubulin aggregation; and taxol, which promotes microtubule formation and stability, all significantly diminished VEC invasion (Fig. 2h)[49,55]. These results indicate that similar to the bladder cell invasion pathway, invasion of VECs requires both actin and microtubule remodeling.

Previous work demonstrated that bladder cell invasion is dependent on the histone deacetylase 6 enzyme (HDAC6), which controls α-tubulin acetylation and consequently microtubule stability[49,56]. We probed at the importance of acetylation in VEC invasion through a pharmacological approach with the gentamicin protection assay. Butyrate, which inhibits HDACs except for HDAC6 and HDAC10, did not inhibit VEC invasion (Fig. 2i), suggesting that one or both of HDAC6 or 10 may be involved in mediating UPEC internalization by VECs; however, trichostatin A, which inhibits HDAC6 and HDAC10, had no impact on VEC invasion (Fig. 2i). Taken together, these experiments (Fig. 2i) suggests no apparent role for the tested histone acetylases in mediating UPEC entry into VEC[56]. Furthermore, nicotinamide, an inhibitor of the deacetylase Sirt2 (Fig. 2i) also had no impact on invasion. None of the inhibitors used in these experiments exerted a significant effect on *E. coli* adherence (Supplementary Fig. 1c–g). Combined, these data indicate a distinct mode of internalization deployed on the vaginal epithelial interface.

Scanning electron microscopy (SEM) of infected VK2 E6/E7 cells at different timepoints revealed UTI89 cells that appear to be heavily decorated with pili adhering to the VEC surface as early as 30 min post-infection (Fig. 2j). Conversely, the Δ*fimA-H* mutant appeared smoother (Fig. 2j) and less tightly associated with the VEC at these early time points (Fig. 2j). SEM images of the 30 and 60 min post-infection samples revealed wild-type UPEC cells regularly found enveloped by the VEC microstructures (Fig. 2j),

contrary to Δ*fimA-H* bacteria that remained loosely associated with the VK2 E6/E7 cell and were not enveloped by membranes (Fig. 2j). Taken together, these results suggest the early adherence and invasion into VK2 E6/E7 proceeds through a zipper-like mechanism and is mediated—at least in part—by type 1 pili.

**The vagina serves as a reservoir for *E. coli*.** To determine the extent of vaginal colonization by *E. coli* after the onset of UTI, we next used a common murine acute cystitis model[57]. C3H/HeN female mice were trans-urethrally inoculated with UPEC strain UTI89, and bacterial burdens in the bladder, kidneys, and vagina were measured 24 h post trans-urethral inoculation. As expected, UTI89 colonized the bladder and kidney tissues at typical levels for this model (Fig. 3a)[57]. The abundance of UPEC titers within the vagina was at levels comparable to those of the bladder (Fig. 3a). Subsequent, immunofluorescence analysis of excised vaginal epithelia isolated from three mice, revealed the presence of both adherent and intracellular *E. coli* (Fig. 3b, Supplementary Movie 1), similar to what was observed in the in vitro model (Figs. 1, 2).

We next assessed the ability of UPEC to persist within the vaginal niche for an extended period of time following acute UTI. For these studies, mice were inoculated trans-urethrally and followed over 4 weeks using urinalysis, as previously described[58]. Though fluctuating, all mice demonstrated bacteriuria over the infection period (Fig. 3c). At the time of sacrifice, the reproductive tract contained the highest levels of bacteria among the excised organs (Fig. 3d). Critically, the mice with bladder bacterial burdens at or near the limit of detection were among those with the highest bacterial burdens in the vaginal, cervical, and uterine horn tissues (Fig. 3c, d). Notably, mice with low-level bacteriuria at any of the tested time points were moved to separate cages from those mice that sustained high levels of bacteriuria. Strikingly, mice with bacteriuria and low bacterial titers in the bladder at the time of humane euthanasia sustained high levels of bacteria in the genital tract (Fig. 3c, d pink lines). These observations suggest that in mice with resolved UTI, based upon bladder counts, a vaginal reservoir persists that may sustain the observed bacteriuria. Immunofluorescence microscopy on excised murine vaginal epithelia revealed adherent UPEC onto murine VECs (Fig. 3e), similar to what was observed during acute infection (Fig. 3b). Additionally, small pod-like groups were observed within the murine VECs (Fig. 3e and Supplementary Movie 2), along with small filaments associated with the epithelial surfaces (Fig. 3e). SEM revealed piliated UPEC adhering to VECs (Fig. 3f) as well as UPEC cells undergoing invagination into the VEC, which is indicative of zipper-like invasion (Fig. 2j, f). These results indicate that UPEC invades murine VECs during the

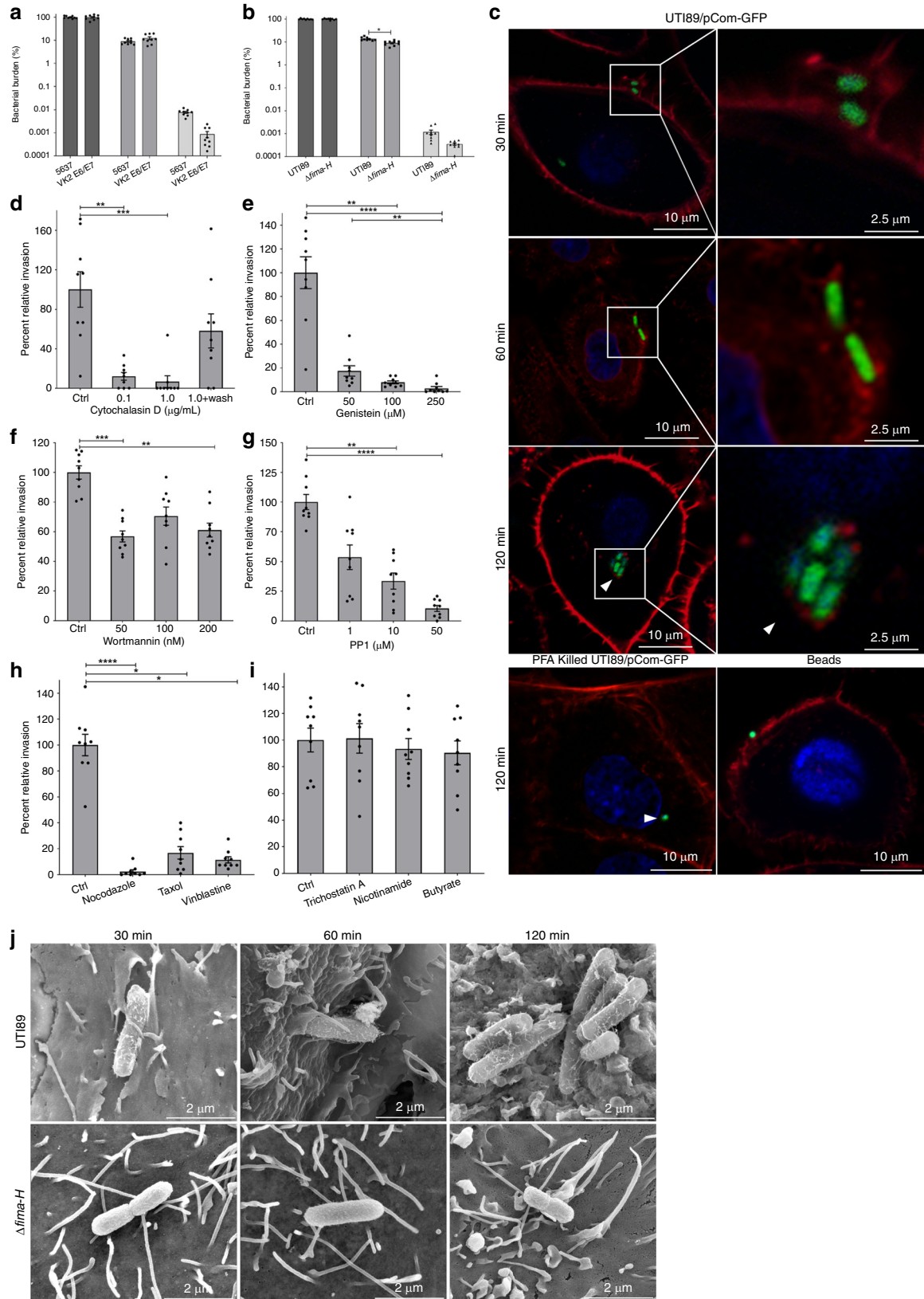

course of a UTI, and vaginal colonization may be associated with persistent bacteriuria in mice with resolved bladder infection.

The moderate reduction of Δ*fimA-H* adherence to VK2 E6/E7 cells suggest that type 1 pili may play a part in colonization of the vaginal space. To elucidate this, we assessed the colonization fitness of the Δ*fimA-H* mutant in the urinary and reproductive

tracts during acute (24-h) and chronic (28-day) infection models. As anticipated, the Δ*fimA-H* strain was significantly attenuated in its ability to establish and sustain bladder infection (Fig. 3g, h). In stark contrast to the urinary tract, we did not observe differences in the total or intracellular bacterial burdens of the two strains in the vagina, cervix, and uterine horns (Fig. 3g, h). Combined, these

**Fig. 2 UPEC invades VECs through a zipper-like mechanism. a–h** Adherence and invasion assays were performed at a MOI of 5:1 for 2 h. Adherent (medium gray bars) and intracellular (light gray bars) bacterial populations are relative to the average of the total population of bacteria (dark gray bars) within the well. **a** Adherence and invasion assay comparison of UTI89 and Δ*fimA-H* strains on VK2 E6/E7 cells and **b** comparison of UTI89 on 5637 bladder and VK2 E2/E7 vaginal cell lines. Data are the mean of nine independent experiments with error bars representing s.e.m. For statistical analysis two-way ANOVA with Sidak's multiple comparisons test ($*P = 0.0402$). **c** Adherent and invasive UTI89/pCom-GFP associated with VK2 E6/E7 with stained by phalloidin conjugated with tetramethylrhodamine, for F-actin and ToPro-3. Red bundles specifically around bacteria indicated actin polymerization occurs prior to invasion. Experimental controls include PFA-killed UTI89/pCom-GFP and latex beads. **d–i** Invasion as determined by gentamicin-based invasion assay with UTI89 with the indicated drug. **d** F-actin inhibitor, cytochalasin D, reversibly inhibited UTI89 invasion into VECs. **e** Genistein, a specific inhibitor of protein tyrosine kinases inhibited UTI89 VEC invasion. **f** Wortmannin, a potent inhibitor of PI3K, minorly reduced UTI89 VEC invasion. **g** PP1, an inhibitor of Src-family kinases, inhibited VEC invasion in a dose-dependent manner. **h** Three different inhibitors of microtubules: nocodazole, taxol, and vinblastine diminished UTI89 VEC invasion. **i** Three histone deacetylase inhibitors trichostatin A, nicotinamide, and butyrate do not prevent UTI89 from invading VECs. Percent invasion is relative to the control group (DMSO). Data are representative of the mean of nine independent experiments with error bars representing s.e.m. **d–h** For statistical analysis non-parametric Kruskal–Wallis with two-sided Dunn's post-hoc test ($*P < 0.05$, $**P < 0.01$, $***P < 0.001$, $****P < 0.0001$). **j** SEM images of UTI89 and Δ*fimA-H*, type 1 pili mutant strain, interacting with the surface of VK2 E6/E7 cells. Microfolds and microvilli are typical of VECs. SEM images of the subtle envelopment of UTI89 by VEC membranes is consistent with a zipper-like model of invasion. Images are representative of three independent experiments.

experiments demonstrate that *E. coli* vaginal colonization is only partly reliant—in the early stages of infection—on the type 1 pili and that other mechanisms mediate invasion and persistence in the reproductive tract.

Thus far, our data indicate that *E. coli* murine vaginal colonization occurs after a UTI and that in mice with resolved bladder infection, vaginal colonization, and potentially VICs contribute to bacteriuria (Fig. 3g, h). To directly test whether *E. coli* can transverse the perineal space between the urinary and reproductive tracts, we vaginally inoculated mice and tracked urine bacterial burdens at distinct timepoints over a 28-day period. Vaginal inoculation of UTI89 in C3H/HeN mice resulted in bacteriuria and colonization of the urinary and reproductive tracts (Fig. 3i, j). During the course of urinalysis, all mice had bacteriuria; those with titers above $1 \times 10^4$ CFUs/mL of urine were placed into new cages, separate from those mice that had low-level bacteriuria. At the time of sacrifice, 28 days post-infection no bladder intracellular populations were detected, however bacterial titers burdens were in the kidneys (Fig. 3i). As with the trans-urethral inoculation cohort (Fig. 3c, d), vaginally inoculated mice with the lowest level of urinary tract colonization exhibited the highest bacterial titers within the vagina, cervix, and uterine horns (Fig. 3i). This experiment demonstrates the ability of vaginal *E. coli* to travel to the urinary tract and cause bacteriuria.

To test the relevance of these findings to human infection, we collected vaginal swabs from women with a history of rUTIs in search of UPEC-associated VECs. Using an *E. coli*-specific antibody on these samples, along with antibodies for uroplakin III and cytokeratin 13 to differentiate bladder epithelial cells from VECs[59,60], we observed intracellular *E. coli* (Fig. 4). Generated *XY*, *YZ*, and *ZX* planes (Fig. 4a–c), as well as surface projection images (Fig. 4e and Supplementary Movie 3), using a Z-stack of the VEC confirmed the localization of *E. coli* within the VEC (Fig. 4). This pilot study demonstrates the ability of *E. coli* to invade vaginal cells and establish VICs in women.

## Discussion

In 1910, Dr. Box and colleagues noted that UTIs arose from *E. coli* ascending the urethra to the bladder[61]. While the vagina has been considered a reservoir for uropathogens for over a century, few studies have dissected UPEC interactions with the host reproductive tract. In this work, we provide critical insights into the ability of UPEC to move between the urinary and reproductive tracts. Furthermore, we demonstrate that UPEC can be internalized by the vaginal epithelium, a step previously unrecognized in the pathogenic cascade that contributes to infection.

A critical first step in the bacteria–epithelial cell interaction is contact. UPEC has a multitude of adhesive appendages at its disposal to engage different receptors in mammalian hosts. For example, the UPEC pangenome harbors a staggering 458 operons that code for chaperone-usher pathway pili[50]. A UPEC strain can carry up to 16 different such gene clusters, thus exhibiting the ability to elaborate multiple fibers tipped with adhesins that have stereo-chemical specificity for different host receptors[17,46]. The *fim* operon that codes for type 1 pili is one example, which has been shown to be critical for infection establishment in the bladder[25–28]. Previous studies elucidated that adherence to VECs is also mediated by type 1 pili[21]. Congruent with the previous studies, we found that type 1 pili do contribute to UPEC adherence. However, our expanded murine infection studies demonstrate that type 1 pili are not necessary for colonizing the vagina or other reproductive organs, suggesting a more critical role for other fibers or adhesive moieties in mediating reproductive tract colonization. In addition, we demonstrate that UPEC can invade VECs through a zipper-like mechanism that is different from that employed during interaction with the bladder epithelium. With the zipper mechanism invading bacteria manipulate host cells to begin endocytosis by binding to host receptors and initiating an intracellular signaling cascade of kinases that promote actin and microtubule rearrangement and pathogen uptake[62]. *E. coli* bladder cell invasion is type 1 pili dependent and requires host PI3K and HDAC6; whereas, our findings show that type 1 pili are dispensable in VEC invasion and requires host Src kinase and not HDAC activity (Fig. 2b–i). The distinction between these different paths for invasion likely arise from adhesive appendages other than type 1 pili activating VEC kinases and adaptor protein regulation of the cytoskeletal network. To remain in accordance with the nomenclature of UPEC bladder colonization stages, we refer to the invading UPEC as VICs (Fig. 4f). We propose that VICs represent a previously unrecognized step in UTI pathogenesis that promotes the longevity of UPEC within the host and subsequently increases the chance of UPEC ascension (Fig. 4f). Our murine models suggest that *E. coli* in VICs are safe from extracellular stressors, such as antibiotics like gentamicin, further confounding solutions for patients suffering from rUTI (Fig. 3). Our murine infection experiments underscore the vagina as a reservoir for *E. coli* that can ascend to the urinary tract to cause infection. Given the prevalence and recurrence rates of UTI, understanding the role of VICs will be a critical factor that influences the success of future therapeutic strategies to manage UTIs.

Notably, during the time of review of this manuscript, O'Brien et al. reported the ability of *E. coli* to colonize the murine vagina

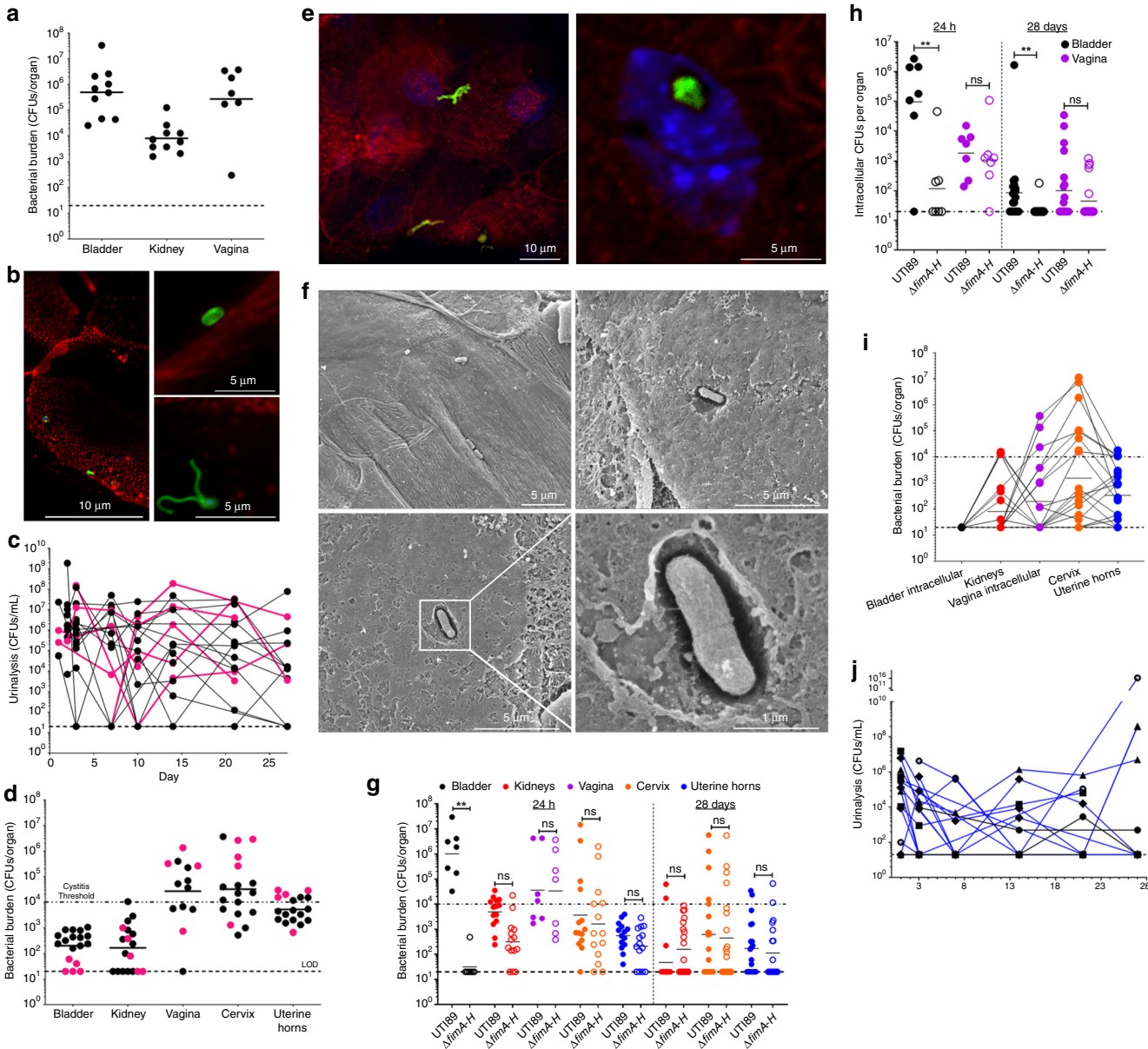

**Fig. 3 UPEC invades VECs during colonization of the reproductive tract during and after UTI in murine models.** C3H/HeN female mice were trans-urethrally infected with UTI89 in **a**, **b**, **g**, **h** acute (24-h infection) and **c**–**h** chronic (for 28 days) UTI models. **a** Bacterial burdens of an acute UTI in the bladder, kidneys, and vaginal membranes from mice with an acute infection, and **b** representative images of immunofluorescence of vaginas from three mice. Chronically infected mice **c**, urine titers **d**, bacterial burden of organs from urinary and reproductive tract. Pink highlight designates mice at or near the limit of detection (LOD) in the **d**, bladder and abundant titers in the reproductive tract with **c**, bacteriuria. **e** immunofluorescence and **f** SEM performed on vaginal tissue of six chronic mice. Immunofluorescence was performed with α-*E. coli* antibody (green) and rWGA (red) and ToPro-3 (blue) staining showing adherent and intracellular *E. coli*. SEM revealed membrane invagination around *E. coli*. The Δ*fimA-H* strain is attenuated in **g** total bladder burden and **h** bladder cell invasion within acute and chronic murine models; whereas, UTI89 and Δ*fimA-H* strains colonize the **g** vagina, cervix, and uterine horns, as well as, **h** invade vaginal cells to similar extent. **i** and **j** Mice vaginally inoculated with UTI89 and followed for 28 days (*n* = 22). **j** Mice with low urinary tract titers also had the highest level of *E. coli* in the vagina, cervix, and uterine horns. **i** horizontal line represents the geometric mean of the organ titers. **j** Vaginal inoculation of UTI89 resulted in individual mice with bacteriuria (blue lines). When urine titers reached ≥1 × 10⁴ CFUs/mL mice were separated into a new cage. To assesses organ colonization, samples were homogenized for bacterial titers and to minimize use of animals some samples were set aside for microscopy. Dashed lines represent LOD, and dotted line represents threshold for cystitis. **g** For statistical analysis two-way ANOVA with Sidak's multiple comparisons test (**$P$ = 0.0013). **h** For statistical analysis non-parametric Kruskal–Wallis with two-sided Dunn's post-hoc test (** for 24-h $P$ = 0.0062; for 28 days $P$ = 0.0078).

and ascend to the uterine horns, consistent with our observations of *E. coli* colonizing the cervix and uterine horns[63]. Further, our experiments demonstrate that *E. coli* can transverse from the urinary tract to the vagina following a urinary tract infection, where it possibly can persist via engaging the extracellular and intracellular environments. In contrast, we show that in vaginally inoculated mice *E. coli* is able to move from the vagina into the urinary tract and demonstrate bacteriuria (Fig. 3). The bacteriuria of vaginally inoculated mice with no bladder and high vaginal titers is not unlike the clinical cases of ASB, specified by

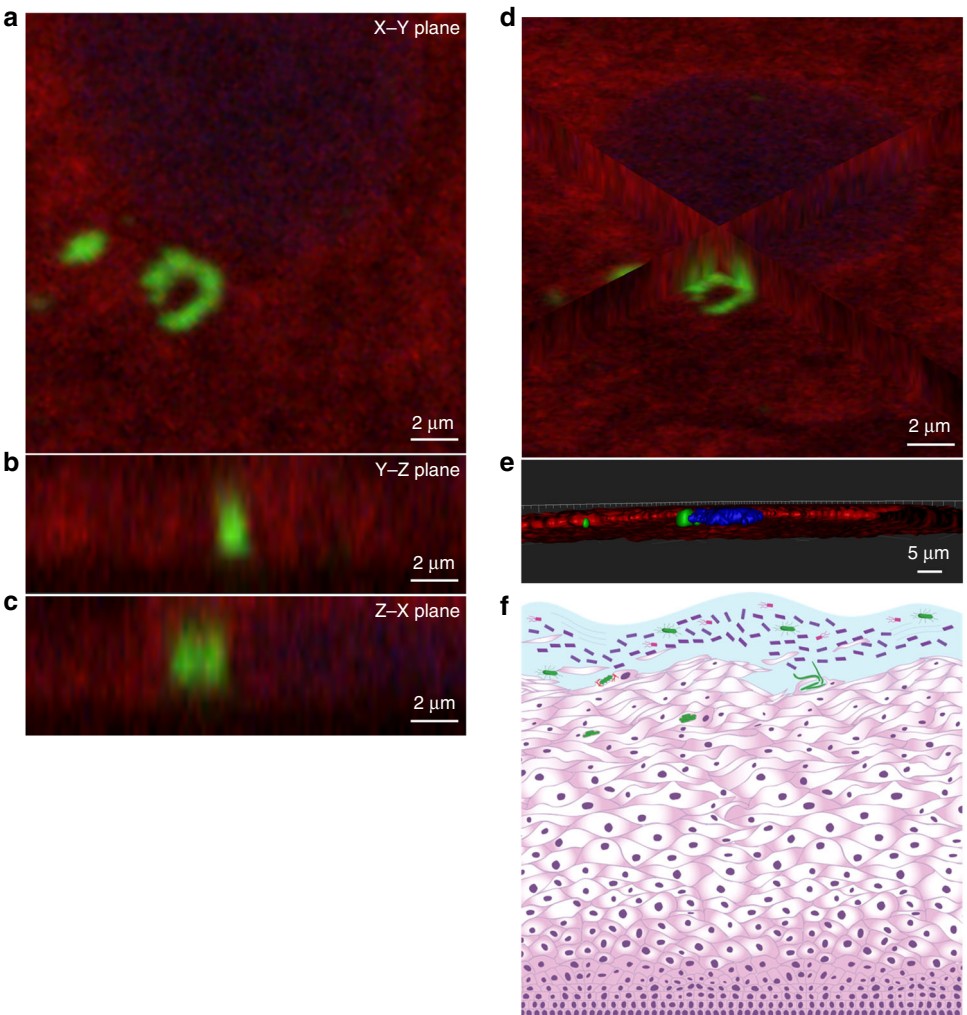

**Fig. 4 UPEC invades the VECs of women with a history of rUTIs.** For this pilot study, immunofluorescence microscopy was used to search for *E. coli* within VEC collected by vaginal swabbing of four women with a history of rUTIs. **a–d** For immunofluorescence, samples were stained with an α-*E. coli* (green), cytokeratin 13 (red), and α-uroplakin III (yellow) and ToPro-3 (blue). **a** The *XY*-plane image from a Z-stack series of a VEC with *E. coli* surrounded by the VEC-specific cytokeratin 13. Ortho views of **b** *YZ*-plane and **c** *XZ*-plane reconstructed from a Z-stack series images showing *E. coli* surrounded by cytokeratin 13 within a VEC near the nucleus. **d** Ortho-view of images mid-way through the VEC demonstrating the internal localization of invading *E. coli* on all three dimensions. **e** Surface reconstructions of *E. coli*, VEC cytokeratin 13, uroplakin III, and nucleus that shows that the nucleus and *E. coli* are within the cytokeratin 13 of the VEC. Imaris software was used for image analysis. **f** Illustration depicting invasion of the vaginal epithelium by UPEC (green bacteria).

quantitative urine titers of $\leq 10^5$ CFUs/mL, that confound the use of antibiotics[64]. Our future research on vaginal *E. coli* colonization may help in the proper diagnosis of UTIs and distinguish ASB from other cases.

## Methods

**Ethics statement**. All mouse experiments were approved by the VUMC Institutional Animal Care and Use Committee (IACUC) (protocol numbers M/12/191, M1500017-01, and M1800101-00). All experiments were conducted in accordance with the guidelines of the National Institute of Health and IACUC at VUMC. Collection and analysis of human samples received internal review board (IRB #180973) approval through the VUMC Human Research Protections Program.

**Cell lines, bacterial strain, and growth conditions**. *E. coli* strains and plasmid used in this study are listed in Supplementary Table 1. *E. coli* were grown at 37 °C in lysogeny broth (LB) on 1.5% LB agar plates supplemented with the appropriate antibiotics: 100 μg/mL ampicillin or 50 μg/mL kanamycin. pCom-GFP was transformed into UTI89 by electroporation to generate strain UTI89/pCom-GFP. For inoculation of VK2 E6/E7 cells and C3H/HeN mice, a colony of the specified UPEC strain was inoculated into 5 mL of LB and cultured at 37 °C while shaking for 5 h. Following, UPEC was consecutively sub-cultured (1:1000 dilution) twice through 18–24 h, 10 mL, static LB cultures at 37 °C to induce type 1 pili production[43]. Prior

to inoculation of VECs and mice, UPEC was normalized to ~$2 \times 10^8$ colony-forming units (CFU)/mL in PBS[58]. The human VEC line VK2 E6/E7 (ATCC CRL-2616) were cultured in keratinocyte-serum-free media (KSFM) (Life Technologies Co., Grand Island, NY) supplemented with 0.1 ng/mL human recombinant epidermal growth factor, 0.05 mg/mL bovine pituitary extract, and 0.4 mM calcium chloride[65]. 5637 (ATCC HTB-9) bladder cells were grown in RPMI 1640 (Life Technologies Co., Grand Island, NY) media supplemented with 10% heat-inactivated fetal bovine serum. VK2 E6/E7 and 5637 cells were grown at 37 °C and 5% $CO_2$.

**Sample collection**. For this pilot study, specimens were collected from four different women of 18 years of age or older with a history of prior UTI or rUTI, defined as three or more UTIs in the past 12 months. Participants were enrolled with their informed consent from the Urology Clinic, within The Vanderbilt Clinic located at the VUMC. Exclusion criteria included presence of indwelling bladder catheter or use of intermittent catheterization, history of urinary diversions or major bladder reconstruction with incorporation of intestinal segments, history of bladder or vaginal cancer, prior or ongoing receipt of chemotherapy, or major medical condition that in the opinion of the investigator would place the subject or study at risk. Upon study enrollment, participants completed a questionnaire about relevant clinical data including information on their current and previous UTIs and hormone menopause status. During routine vaginal/pelvic examination during the patients visit to the clinic, a physician collected a vaginal swab. Biospecimens were

transported to the laboratory for furthering processing and analysis within one hour. Vaginal swabs were suspended and vortexed in PBS to remove cells from the swab. Following, collected samples were centrifuged at $600 \times g$ for 5 min to pellet cells. Pelleted cells were re-suspended in PBS and split into equal aliquots and fixed for microscopy.

**Adherence and invasion assay.** Adherence and invasion assays were performed as essentially described elsewhere[25]. VECs were seeded into 24-well plates and grown to ~90–95% confluency. Where indicated cytochalasin D (0.1 and 1 µg/mL) was applied to VEC for 30 min prior to infection and washed away for 30 min. Where indicated, cells were treated for 45 min with the inhibitors, genistein (50, 100, and 250 µM), wortmannin (50, 100, 200 nM), PP1 (1, 10, 50 µM), nocodazole (66 µM), taxol (40 µM), vinblastine (40 µM), trichostatin A (300 nM), butyrate (5 mM), and nicotinamide (5 mM) prior to infection for 120 min. The addition of UPEC or the above-mentioned drugs had no statistically significant impact on VEC viability during these experiments as previously described with bladder epithelial cells (Supplementary data Fig. 1h)[25,45]. On the day of the experiment, 960 µL of fresh KSFM was applied to the cells. Prior to infection, the number of VECs per well was enumerated to calculate the volume of UPEC suspension (<20 µL) to be added. The UPEC inoculum was enumerated by serial dilution and spot plating on LB agar plates to ensure the proper MOI. The appropriate volume for the specified MOI of UPEC inoculum was added to three sets of triplicate wells. Centrifugation at $600 \times g$ for 5 min was used to facilitate and synchronize contact between UPEC and VECs. Unless indicated otherwise, plates were incubated for 2 h at 37 °C and 5% $CO_2$. To enumerate the bacterial burden, 20 µL of 5% Triton X-100 was added per well when required for VEC lysis. The cells of one set of wells was lysed to determine the total number of extracellular or intracellular bacteria. The second set was washed three times with 500 µL of PBS, lysed, and enumerated. Bacteria not washed away within this set are considered to be adherent. The final set of wells was washed alongside the adherent set; additionally, these wells were incubated for 2 h with fresh KSFM with 100 µg/mL of gentamicin (Life Technologies Co., Grand Island, NY). Following, VECs were washed twice with 1 mL of PBS, lysed, and bacterial titers were enumerated. Percent bacterial adherence and invasion were calculated as a percent of the total number of bacteria.

**Murine UTI models.** C3H/HeN female mice (Envigo) at 7–8 weeks old were trans-urethrally inoculated with $10^7$ CFUs of *E. coli* in 50 µL of PBS[58]. At the time of humane euthanasia (24 h for acute infection and 28 days for chronic infection), organs were aseptically excised and homogenized in 1 mL PBS. To determine intracellular bladder and vaginal *E. coli* titer, the organs were bisected, washed three times in 1 mL PBS, washed in 1 mL of 100 µg/mL of gentamicin for ~2 h, and washed two more times in 1 mL PBS prior to homogenization in 0.1% Triton-X 100. Homogenates were serially diluted and spot plated for bacterial enumeration on LB plates. For urinalysis, urine released from the urethra—located anterior to the vaginal introitus—was collected into sterile 1.5 mL microcentrifuge tubes to determine urine titers[58]. Vaginal membranes for microscopy were bisected and one half was washed and fixed for SEM with 2.5% glutaraldehyde in 100 mM sodium cacodylate (Electron Microscopy Sciences, Hatfield, PA). The second half was washed and fixed for immunofluorescence in 3.4% PFA (Electron Microscopy Sciences Hatfield, PA) in PBS.

**Murine vaginal inoculation.** Similar to previously reported models for murine vaginal inoculation models, C3H/HeN, 7–8-week-old mice (Envigo) were estrogenized by intraperitoneal injection of filter sterilized 0.5 mg β-estradiol in 100 µL sesame oil 3 and 1 day prior to vaginal inoculation to synchronize mice[66–69]. Anaesthetized mice were vaginally inoculated with $10^7$ CFUs of strain UTI89 in 20 µL of PBS. After human euthanasia, organs were aseptically excised and homogenized as described above as done for tissues collected in the murine UTI models. Urinalysis was conducted for vaginally inoculated mice as described for the murine UTI models[58].

**SEM and TEM microscopy.** SEM and TEM were performed to confirm localization and view cell–cell interactions[70]. For TEM, VK2 E6/E7 cells were grown in six-well tissue culture plates and infected with strain UTI89 at the indicated MOI as described for the adherence and invasion assay. At the indicated time point cells were washed, and fixed in 2.5% glutaraldehyde in 100 mM sodium cacodylate for 30 min at room temperature. Specimens were placed on glow-discharged formvar/carbon-coated copper grids and stained with 1% uranyl acetate for 90 s. A Philips/FEI T-12 transmission electron microscope was used for sample analysis. For SEM, VK2 E6/E7 cells were grown on glass cover slips in 12-well plates, infected with the bacteria for 2 h and fixed with 2.5% glutaraldehyde in 100 mM sodium cacodylate for 30 min at room temperature. Samples were critical point dried and placed onto aluminum stubs prior to sputter coating with gold-palladium. Images were collected for each sample and representative images were taken with a FEI Quanta 250 field-emission gun scanning electron microscope (Field Electron and Ion Company, Hilllsboro, OR). Images were taken from a minimum of three biological replicates.

**Fluorescence microscopy.** Fluorescence microscopy was performed as described elsewhere[25,59]. VK2 E6/E7 cells grown in six-well plates on glass slides and infected with UTI89 at a MOI of 5 as described for the adherence and invasion assay. Where indicated, PFA-killed bacteria were treated with 3.4% PFA for 90 min prior to addition to VECs. Where indicated, $2 \times 10^6$ fluorescently labeled, yellow-green, 1.0 µm, latex beads (Sigma-Aldrich, St. Louis, MO) were added per well. Samples were washed three times with PBS and fixed with 3.4% PFA in PBS at room temperature for 30 min. Samples were permeabilized with 0.1% Triton X-100 for 20 min and washed with PBS three times. Where indicated, VECs were stained with wheat germ agglutinin (r-WGA) (1:500 dilution), to outline cell membranes or phalloidin (1:40 dilution), to stain F-actin, conjugated with tetra-methylrhodamine (Molecular Probes, Eugene, OR). For immunofluorescence, samples were permeabilized, blocked with 2% BSA overnight and washed once with PBS before proceeding to staining. Primary antibodies were applied in PBS with 0.1% BSA overnight at room 4 °C with gentle rocking. Primary antibodies used in this study included rabbit α-*E. coli* antibody (1:1000) (US Biologicals) that is specific for *E. coli* and non-reactive with vaginal microbiota, such as *Lactobacillus crispatus* (Supplementary Fig. 1a), mouse α-uroplakin III antibody (1:1000) (Abcam), goat α-cytokeratin 13 antibody (1:500) (Abcam)[71]. Samples were washed once with PBS prior to secondary antibody application. Secondary antibodies were applied in PBS with 0.1% BSA for 1 h at room temperature with gentle shaking. Secondary antibodies (dilution 1:1000) used in this study were donkey α-goat, α-rabbit, and α-mouse IgG conjugated with Alexa Fluor 546, 488 or 647, or 594, respectively (ThermoFisher Scientific). ToPro3 (1:1000 dilution) (Molecular Probes, Eugene, OR), a DNA-specific stain, was used for counter staining for 20 min. All samples were washed a final three times with PBS and mounted using ProLong Diamond (Molecular Probes, Eugene, OR). Images were acquired with a LSM 710 META Inverted (Zeiss International, Oberkochen Germany) with plan-apochromat ×63/1.4 immersion oil objective. Post-acquisition analysis was performed using Zen (Zeiss International, Oberkochen Germany) and Imaris (Bitplane, Zürich Switzerland) software. Images were cropped in Photoshop CC (Adobe, San Jose, CA) for display into figures.

**Statistical analysis.** GraphPad Prism was used for all statistical analyses performed using the most appropriate test. Statistical test details are listed along with error bars and statistical significance cutoffs within figure legends.

**Reporting summary.** Further information on research design is available in the Nature Research Reporting Summary linked to this article.

## Data availability
The authors declare that all data supporting the findings of this study are available within this manuscript and its supplementary information files. The source data underlying Figs. 1a, b, f, 2a, b, d–i, 3a, c, d, g–j; and Supplementary Fig. 1b–j are provided as a Source Data file. Source data are provided with this paper.

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

## Acknowledgements

We would like to thank Drs. J.W. Brock and D.B. Clayton in the VUMC Department of Urology for their advice during the initial stages of this project. We also thank Dr. D.M. Aronoff and the Vanderbilt Preventing Adverse Pregnancy Outcomes and Prematurity Initiative group for thoughtful discussions concerning the female reproductive tract. We thank Drs. J.E. Schmitz and C.W. Stratton at the VUMC Clinical Microbiology Laboratory for collection of clinical laboratory strains provided through the microVU biobank under the Initiative for Personalized Microbial Discovery and Innovation that is supported through Vanderbilt University's Trans-Institutional Programs and for providing the L. crispatus. We are thankful to Dr. A.R. Eberly for technical assistance and training for initial murine UTI experiments. Sample preparation for SEM and TEM was performed in part through the use of the Vanderbilt Cell Imaging Shared Resource (CISR) (supported by NIH grants CA68485, DK20593, DK58404, DK59637, and EY08126). Microscopes and software utilized for imaging and image analysis were provided through CISR. This reported research was supported by grants from the NIH AI107052-01A1 and DK123967-01 (M.H.), K23DK103910 (W.S.R.), and T32 5T32AI112541-05 (M.A.W.). Research in this publication was supported, in part, by the National Institute of General Medical Science of the NIH under award number T32 GM007569 (J.R.B.) and March of Dimes 6-FY17-295 to David Aronoff (J.R.B.).

## Author contributions

J.R.B. and M.H. conceived and designed the experiments. J.R.B. performed adherence and invasion assays for UTI89 and ΔfimA-H on VECs. J.R.B., T.L.D., C.J.B., and T.R. performed adherence and invasion assays with various clinical UPEC isolates. Adherence and invasion experiments with different drugs were performed by J.R.B. and T.L.D. J.R.B. performed fluorescence microscopy, SEM, and TEM experiments. CISR facility performed sample preparation for SEM and TEM. J.R.B. acquired images for SEM and TEM. Mouse experiments were performed by J.R.B., C.J.B., M.A.W., and M.H. Human clinical samples were collected by W.S.R. and processed by J.R.B. and T.L.D. Data analysis was performed by J.R.B., T.L.D., and M.H. The manuscript was written by J.R.B. with assistance from T.L.D. and edited by J.R.B., C.J.B., and M.H.

## Competing interests

The authors declare no competing interests.
