## [Peer Review File · Nature Communications]

Reviewers' comments:

Reviewer #1 (Remarks to the Author):

In this nicely written paper, Brannon et al describe the ability of UPEC to invade vaginal epithelial cells. They show that many UPEC strains have this ability, and that the invasion process is in part mediated by type 1 pili. Data from a mouse UTI model and from human vaginal washes suggest that UPEC invasion of vaginal epithelial cells (VECS) does occur during the course of a UTI. Overall, the work is fairly solid and should be of great interest to the field and those interested in host-bacteria interactions. However, the main conclusion that intracellular reservoirs of UPEC within VECs may promote UTI and bacterial persistence within the host should be better substantiated (see below).

Major issues:

1. The authors propose that Vaginal Intracellular Communities (VICs) "represent a previously unrecognized step in UTI pathogenesis that promotes the longevity of UPEC within the host and subsequently increases the chance of UPEC ascension." The mouse data presented in Fig 3 support this possibility, but could be strengthened considerably. Gentamicin protection assays with reproductive tissues isolated from infected mice could help confirm that the persistent bacteria are intracellular, as the authors' model suggests.

2. If UPEC is inoculated directly into the vagina, will the bacteria persist like they do in the UTI experiments (Fig 3)? Or does vaginal persistence of UPEC require re-seeding from the urinary tract? Alternatively, does UPEC disseminate into the urinary tract from the genital tract if the vagina is inoculated? Answers to these questions would help test the authors' model and strengthen the conclusions considerably.

3. The images indicating intracellular E. coli within vaginal cells from patients is potentially very interesting, but additional controls would help validate the findings (Fig 4). Since the vaginal mucosa has a fairly rich microbiota, can the authors confirm that what they are observing is actually UPEC and not other bacteria that cross-react with the anti-E. coli antibody? Use of additional antibodies with different specificities would help, and ELISAs or flow could be used to check for potential cross-reactivity with members of the vaginal microbiota. Similar controls could help help strengthen the image analysis of infected mice, as presented in Fig 3.

Other issues:

4. In the SEM images in Fig 3f it is difficult to appreciate invading bacteria or membrane zippering around the bacteria, as described in the text (page 6). Higher magnification images might help. As is, it is difficult to tell if the bacteria are just in crevices within the tissue, or are they being enveloped. More concerning, based on the scale bars, the bacteria in the SEM are much smaller length-wise than typical E. coli (maybe only 0.5 microns, based on scale bars?). This should be addressed.

5. Fig. 1 concerns. The presentation of similar types of data using different types of graphs in a, b, and c is at first a little confusing. Might be better to stick with one style. Fig 1c is referenced in the text after Fig 1d and e. The text or figure should be modified so that the data in the figure is presented more sequentially.

ASB is not defined for Fig 1c, Table S1 or elsewhere.

For Fig 1c, the authors state that 'These strains exhibited similar adherence levels and all exhibited the ability to become internalized.' However, 4 of the tested strains are at or below the limit of detection (LOD). This statement should be qualified and LOD should be defined in legend or text.

Arrows in Fig 1d indicate "intracellular bacteria that appeared enclosed in vacuoles". The arrows should be noted in the figure legend. Based on these images, it is not clear that the bacteria are enclosed within vacuoles. It might be helpful to differentially label the intra- and extracellular bacteria to better confirm localization within the VECs (e.g. use anti-E. coli antibody without permeabilizing the host cells).

6. For Fig 3a, b, d, e, and f, times points assayed should be indicated in the legend (not just in text).

7. Line 26 (opening sentence) - This is a very broad statement, especially because 'reservoirs' can be defined in many different ways both within the host and within the environment. The authors are presumably referring to reservoirs within hosts. This should be clarified and cited.

8. The Legends for Fig 1, 2 and S1 indicate that the data are "representative", but it is not clear if this is what the authors really mean. The graphs should ideally show all of the data from independent replicates, not representative data. Please clarify.

9. Lines 132-134: "SEM revealed piliated UPEC adhering to VECs (Fig. 3f)...". SEM is unlikely to resolve individual pili. How do the authors know these bacteria are piliated?

Reviewer #2 (Remarks to the Author):

This manuscript by Dunigan et al details the ability of uropathogenic E. coli (UPEC) to invade vaginal epithelial cells (VEC), and proposes vaginal intracellular communities (VICs) as a previously unrecognized step in UPEC urinary tract infection (UTI). The authors note that multiple studies have reported vaginal colonization by UPEC during or prior to UTI and the ability of UPEC to adhere to vaginal epithelial cells, although this is the first study to clearly demonstrate invasion of vaginal cells by UPEC. The experimental findings are further strengthened by visualization of intracellular UPEC in VECs from human subjects with recurrent UTIs.

The authors show that during UTI in mice, UPEC colonizes the reproductive tract and persists in this niche even after bladder infection has resolved. They also show that levels of UPEC in urine and the reproductive tract can remain quite high after resolution of bladder infection, and suggest that the vaginal niche contributes to sustained bacteriuria. The novelty of this finding is slightly reduced by a very recent publication in PLoS One regarding UPEC colonization of the murine reproductive track, which proposed that UPEC in urine could lead to vaginal colonization, and that vaginal colonization could also contribute to UPEC titers in urine (PLoS One, 2019 Jul 22;14(7):e0219941). The findings of this PLoS One article warrant discussion in the present manuscript, particularly a comment on whether or not repeated urine collection from mice in this study may have contributed to sustained vaginal colonization in the longer-term infection experiments.

Regarding quantification of VEC adherence and invasion in Figure 1A, 1B, and 2A, it would be useful to have a direct comparison to adherence and invasion of bladder epithelial cells using the same MOI. These panels would also benefit from statistical analysis. Furthermore, how consistent was adherence and invasion between technical and independent replicates? Displaying the standard deviation instead of standard error of the mean may be beneficial to facilitate comparison across MOIs and between the parental strain and the type I pilus mutant.

In Figure 2, the authors show that a type I pilus mutant (*fimA-H*) has decreased VEC adherence and invasion compared to the parental strain, but these capabilities are not completely ablated in

the mutant. Are there other known UPEC receptors that are expressed on both bladder and vaginal epithelial cells that are likely to contribute to Fim-independent attachment?

Considering that type I pilus contributes to VEC invasion, is the fimA-H mutant still capable of colonizing the reproductive tract during urinary tract infection? Infection studies with this mutant could further delineate the importance of vaginal colonization in seeding recurrent UTI.

Regarding visualization of intracellular UPEC in VECs from human subjects with recurrent UTIs, these observations would be greatly strengthened by inclusion of the number of specimens that were screened and the number that had UPEC in VECs. If possible, assessment of samples from subjects without a history of recurrent UTI would also be beneficial, to determine how common E. coli VICs are in otherwise healthy women versus their association with recurrent UTI.

Lines 198-200: this sentence is confusing as written

Line 203: remove the comma between "serial" and "dilution"

Line 205: "as" should be "was"

Line 208: "lyses" should be "lysis"

Line 244: add "were" between "cells" and "grown"

Reviewer #3 (Remarks to the Author):

GENERAL:

This short report documents that UPEC can invade vaginal epithelial cells, with the authors proposing that these cells may represent an intracellular reservoir for UPEC during UTIs. The conclusions are based on studies with a vaginal epithelial cell line, a mouse transurethral infection model, and analyses of cells from patients with recurrent UTIs. The manuscript contains some compelling microscopy data, using both confocal microscopy and SEM, particularly for the in vitro cell line studies (I found the mouse model data on intracellular UPEC in Fig. 3b and 3e to be a little less convincing). The issues highlighted below should be addressed by the authors.

MAJOR:

1. The authors state in their opening paragraph, "Because the intestines and vagina are considered to be the primary reservoirs for UPEC – the primary cause of UTIs – infection of the urinary tract likely occurs through ascension of UPEC from these reservoirs to the urethra (reference 1)". Indeed, there is literature on the vagina as a UPEC reservoir dating back several decades, and there are several published studies demonstrating adherence of UPEC isolates to vaginal epithelial cells (VECs). Thus, the novelty in the current study appears to be restricted to the fact that UPEC can not only adhere to VECs, but also invade and survive within these cells. The findings presented certainly provide evidence of this. However, given that it is now very well-established that UPEC occupies both extracellular and intracellular niches, it is not entirely clear to this reviewer as to how novel and impactful these findings are. In my view, a better case could be made for this in the way that the authors present their findings in the manuscript. In addition, more detailed characterization of the compartment (see below) might ensure that this work has greater influence in the field of UPEC pathogenesis.

2. The authors show evidence of an intracellular niche for UPEC within VECs in Figs. 1-2. More detailed characterization of the compartment (e.g. positivity for specific endolysosomal markers such as LAMP1, PMIDs: 16968784; 19451249) and mode of entry (e.g. role of microtubules and

HDAC6, PMID: 18996840), as per the existing literature on bladder epithelial cells, would be highly desirable. Functional data on specific host factors that support invasion and/or survival in VECs would greatly increase the impact of this work. Without such data, the study could be viewed as somewhat phenomenological, albeit important.

3. Page 6, last 2 lines: What is the "E. coli-specific antibody" that was used? What is the epitope being recognized, and is it certain that this antibody is truly E. coli-specific? This is an important issue if claims are to be made about an intracellular UPEC reservoir in VECs of patients with recurrent UTIs. And how many patient samples are the presented data representative of (this information does not appear to be provided in Fig. 4)?

4. Fig. 2a shows adherence and invasion data for a FimA-H mutant. In order to be able to interpret these data appropriately, the authors should show side-by-side data with the wild type strain in the same experiments (rather than relying on comparisons with the wild type strain in different experiments – Fig. 1a). Otherwise, the data are not directly comparable.

5. In Fig. 1a, intracellular UPEC loads are decreasing with increasing MOIs. This seems surprising. Was cell death monitored? Since UTI89 is a hemolysin-positive strain, is it possible that VECs are being killed during these in vitro infections? This read-out should be assessed, since it could impact on data interpretation.

6. It was not clear to this reviewer if the graphical data for in vitro infections (e.g. Fig. 1a, 1b, 1c, 2a, 2b) were combined from 3 different experiments (i.e. taking mean of each experiment) or if the data are from one experiment showing replicate wells (with similar findings to the presented data being observed in 3 different experiments). The former approach is required for appropriate statistical analysis. Can the authors clarify?

7. Recent studies on a mouse model to study colonization of the vagina by UPEC (PMID: 31329630) and on the impact of the vaginal microbiota on reactivation of UPEC infection in the bladder (PMID: 28358889) would seem to be relevant to this study and could be discussed in the context of the presented findings.

MINOR:

1. Introduction, page 3, lines 51-52: Instead of providing a single review article as a reference for the statement "Multiple studies have reported that UPEC colonizes....", it would be preferable if the authors cited multiple independent primary research papers to support this statement.

2. Page 3, last line: Fig. 1c is described after Fig. 1d. The authors should either alter the text or the order of the panels in the actual figure, so that figure panels are described in order. The same issue applies to Fig. 2b (page 5, first line), which is described after Fig. 2c is presented.

3. Fig. 1 panels and legend: What MOI was used for panel c? Please indicate this (in this panel, the level of invasion for the UTI89 control looks lower than for the data in panels a and b?). Please also provide labels for top and bottom images in panel e. In the legend, "f" should be "e".

4. In Fig. 3, arrows could be used in panel f to highlight specific examples of UPEC undergoing invagination into the VEC.

Reviewer #1 (Remarks to the Author):

In this nicely written paper, Brannon et al describe the ability of UPEC to invade vaginal epithelial cells. They show that many UPEC strains have this ability, and that the invasion process is in part mediated by type 1 pili. Data from a mouse UTI model and from human vaginal washes suggest that UPEC invasion of vaginal epithelial cells (VECS) does occur during the course of a UTI. Overall, the work is fairly solid and should be of great interest to the field and those interested in host-bacteria interactions. However, the main conclusion that intracellular reservoirs of UPEC within VECs may promote UTI and bacterial persistence within the host should be better substantiated (see below).

We are thankful to Reviewer 1 for their time and careful analysis of the manuscript. We agree with them in that the recommend experiments (addressed below) would greatly enhance our findings regarding VEC invasion and VIC formation. We have taken into consideration all comments and have added additional experiments to address all of them. We have adjusted data presentation and included additional mouse experiments including the use of a vaginal inoculation model. The results of all these additional experiments support that *E. coli* establishes VIC during vaginal colonization and demonstrate the vagina serves as an important reservoir for ascending UTIs.

Major issues:

1. The authors propose that Vaginal Intracellular Communities (VICs) “represent a previously unrecognized step in UTI pathogenesis that promotes the longevity of UPEC within the host and subsequently increases the chance of UPEC ascension.” The mouse data presented in Fig 3 support this possibility, but could be strengthened considerably. Gentamicin protection assays with reproductive tissues isolated from infected mice could help confirm that the persistent bacteria are intracellular, as the authors' model suggests.

We thank the reviewer for this suggestion. In the revised manuscript, we provide data of gentamicin-treated organs from acute and chronic UTI models and in a vaginal inoculation model (Fig. 3h and i). Indeed, as the reviewer indicated inclusion of these experiments demonstrated the presence of intracellular reservoirs.

2. If UPEC is inoculated directly into the vagina, will the bacteria persist like they do in the UTI experiments (Fig 3)? Or does vaginal persistence of UPEC require re-seeding from the urinary tract? Alternatively, does UPEC disseminate into the urinary tract from the genital tract if the vagina is inoculated? Answers to these questions would help test the authors' model and strengthen the conclusions considerably.

Guided by the reviewer's suggestion, we evaluated how vaginal inoculation would influence colonization of the bladder. The findings from these new experiments are now incorporated

in the revised figure 3 (Fig. 3i-j). Figure 3i depicts organ colonization at the time of euthanasia, while figure 3j depicts urinalysis over time. These results indicate that vaginal inoculation results in the formation of VICs and colonization of the cervix and uterine horns. Additionally, sustained bacteriuria is observed in several of the vaginally inoculated mice (Fig. 3j), along with colonization of the kidneys. As with the UTI models, mice with high *E. coli* titers within the vagina, cervical and uterine tissues had low urinary tract colonization and still a high level of bacteriuria. These experiments further support our findings that *E. coli* VIC serve as a reservoir.

3. The images indicating intracellular *E. coli* within vaginal cells from patients is potentially very interesting, but additional controls would help validate the findings (Fig 4). Since the vaginal mucosa has a fairly rich microbiota, can the authors confirm that what they are observing is actually UPEC and not other bacteria that cross-react with the anti-*E. coli* antibody? Use of additional antibodies with different specificities would help, and ELISAs or flow could be used to check for potential cross-reactivity with members of the vaginal microbiota. Similar controls could help help strengthen the image analysis of infected mice, as presented in Fig 3.

We appreciate the reviewer's concern for antibody specificity for *E. coli*. *E. coli* antibodies, including the one used in this study, are raised against a mixture of *E. coli* serotypes. The antibody used in this study (US Biologicals catalog number E3500-06 Lot number L8102865) is highly specific for a mixture of *E. coli* O and K antigens, the variety of O and K antigens used to generate this antibody ensure that the broad range of UPEC serotypes seen within the clinical setting can be detected. Nevertheless, we understand the reviewer's concern and have tested the antibody for cross-reactivity with *Lactobacillus crispatus*, the primary constituent of the vaginal microbiota. Figure S1h depicts the results, demonstrating that the *E. coli* antibody is not cross-reactive with *Lactobacillus* antigens.

Other issues:

4. In the SEM images in Fig 3f it is difficult to appreciate invading bacteria or membrane zippering around the bacteria, as described in the text (page 6). Higher magnification images might help. As is, it is difficult to tell if the bacteria are just in crevices within the tissue, or are they being enveloped. More concerning, based on the scale bars, the bacteria in the SEM are much smaller length-wise than typical *E. coli* (maybe only 0.5 microns, based on scale bars?). This should be addressed.

To address the reviewer's concerns here, we have provided higher magnification images of these samples including an image taken at 100,000X magnification (Revised Figure 3, panel f). These images clearly show membrane invagination around *E. coli* between 1-2 microns, typical for *E. coli*.

5. Fig. 1 concerns. The presentation of similar types of data using different types of

graphs in a, b, and c is at first a little confusing. Might be better to stick with one style. Fig 1c is referenced in the text after Fig 1d and e. The text or figure should be modified so that the data in the figure is presented more sequentially.

We thank the reviewer for this recommendation. We have re-ordered the figure panels to incorporate changes made in response to all reviewers' comments throughout the manuscript and have integrated this suggestion as well. Bars within the previous graph were superimposed to conserve space. We have adjusted Fig. 1c to be staggered in the same manner as Fig. 1a and b to show the adherent and intracellular titers of clinical isolates.

ASB is not defined for Fig 1c, Table S1 or elsewhere.

We thank the reviewer for catching this oversight. During the original version of the manuscript, we had removed components of the introduction including the term asymptomatic bacteriuria (ASB). We have expanded the introduction and discussion of the manuscript and in addition defined "ASB" where appropriate on its first appearance.

For Fig 1c, the authors state that 'These strains exhibited similar adherence levels and all exhibited the ability to become internalized.' However, 4 of the tested strains are at or below the limit of detection (LOD). This statement should be qualified and LOD should be defined in legend or text.

We thank the reviewer for bring this to our attention along with their comment about presenting Fig. 1a-c consistently to avoid confusion (comment 5). Consequently, we have addressed this within the figure and adjust the text accordingly with the inclusion of the statement, "exhibited VEC invasion to different levels".

Arrows in Fig 1d indicate "intracellular bacteria that appeared enclosed in vacuoles". The arrows should be noted in the figure legend. Based on these images, it is not clear that the bacteria are enclosed within vacuoles. It might be helpful to differentially label the intra- and extracellular bacteria to better confirm localization within the VECs (e.g. use anti-*E. coli* antibody without permeabilizing the host cells).

As requested by the reviewer, we have provided the details concerning arrows in the figure legend (Fig. 1d). Further, we have provided additional images using strain UT189/pCom-GFP that constitutively expresses GFP in our adherence and invasion assay. Here, samples were stained with an anti-*E. coli* antibody without Triton-X 100 permeabilization (Fig. 1e) as Reviewer 1 suggested. These images clearly show that *E. coli* are well within the membrane of the vaginal cells.

6. For Fig 3a, b, d, e, and f, times points assayed should be indicated in the legend (not just in text).

As per the reviewer's recommendation, we have specified the time points for the acute (24-hour infection) and chronic (28-days) murine infections within the figure legend.

7. Line 26 (opening sentence) - This is a very broad statement, especially because 'reservoirs' can be defined in many different ways both within the host and within the environment. The authors are presumably referring to reservoirs within hosts. This should be clarified and cited.

We agree with the reviewer that reservoirs can be within and outside the host. We think that it is important for the readers from a broader audience to understand the importance of reservoirs in recurrent infections, which are the main topic of this manuscript. We go into further detail of *E. coli* reservoirs within the host in later paragraphs of this manuscript, where pertinent. The sentence in line 26, which was initially the opening statement for the introductory paragraph/abstract in the [Redacted] Letter formatting has now been revised as part of the reformatting of the manuscript for a Nature Communications article. We have also incorporated the dependent clause, "particularly those within the host", to specify the type of reservoir being discussed per the reviewer's request.

8. The Legends for Fig 1, 2 and S1 indicate that the data are "representative", but it is not clear if this is what the authors really mean. The graphs should ideally show all of the data from independent replicates, not representative data. Please clarify.

We agree with the reviewer that wording in this statement should be clarified. Cumulative data from all independent replicates are now presented – along with mean with standard error calculations.

9. Lines 132-134: "SEM revealed piliated UPEC adhering to VECs (Fig. 3f)...". SEM is unlikely to resolve individual pili. How do the authors know these bacteria are piliated?

Although SEM is certainly not as sensitive as TEM in resolving individual pili, it clearly distinguishes between the appearance of the parent strain and the *fim*-deficient isogenic mutant (Figure 2j). We clarify in the revised text and figure 2 legend that the SEM of type 1 pili mutant do not contain the appendages of the parental UTI89 strain for individuals not accustomed to looking at bacterial pili.

Reviewer #2 (Remarks to the Author):

This manuscript by Dunigan et al details the ability of uropathogenic *E. coli* (UPEC) to invade vaginal epithelial cells (VEC), and proposes vaginal intracellular communities (VICs) as a previously unrecognized step in UPEC urinary tract infection (UTI). The authors note that multiple studies have reported vaginal colonization by UPEC during

or prior to UTI and the ability of UPEC to adhere to vaginal epithelial cells, although this is the first study to clearly demonstrate invasion of vaginal cells by UPEC. The experimental findings are further strengthened by visualization of intracellular UPEC in VECs from human subjects with recurrent UTIs.

The authors show that during UTI in mice, UPEC colonizes the reproductive tract and persists in this niche even after bladder infection has resolved. They also show that levels of UPEC in urine and the reproductive tract can remain quite high after resolution of bladder infection, and suggest that the vaginal niche contributes to sustained bacteriuria. The novelty of this finding is slightly reduced by a very recent publication in PLoS One regarding UPEC colonization of the murine reproductive track, which proposed that UPEC in urine could lead to vaginal colonization, and that vaginal colonization could also contribute to UPEC titers in urine (PLoS One, 2019 Jul 22:14(7):e0219941). The findings of this PLoS One article warrant discussion in the present manuscript, particularly a comment on whether or not repeated urine collection from mice in this study may have contributed to sustained vaginal colonization in the longer-term infection experiments.

We are highly appreciative for the reviewer's time commitment to evaluating the quality of our work and their overall positive response to our findings. We agree with reviewer 2 in regards to discussing the manuscript (PLoS One, 2019 Jul 22:14(7):e0219941), which notably was published during the review process of this manuscript. As such, we had not discussed the O'Brien et al., work in the original submission. In the herein revised manuscript, we include discussion of the PLoS One publication. Specifically, in agreement with O'Brien *et al*, our data indicate that prolonged bacteriuria in mice may come from vaginal bacteria. The act of collecting urine does not involve manipulation of the murine genitals, but relies on an individual to catch the urine in a microtube as it falls from the mouse urethra by gravity. The basic anatomy of the female mouse prevents urine from falling upwards into the vaginal introitus. The murine vagina is located posterior to the urethral meatus – as it is in women – and above the urethral meatus. Thus, it is unlikely that the urinalysis resulted in vaginal inoculation. Congruent with the O'Brien et al. observation, vaginal instillation experiments lead to sustained bacteriuria indicating the ability of vaginal bacteria to enter the urinary tract. In this manuscript, we are proposing that the vaginal colonization seen in our experiments and perhaps those published by O'Brien *et al* is sustained by the presence of VICs, which was not tested in the article by O'Brien *et al*. Finally, we have added a section (in the discussion) regarding the contribution of vaginal *E. coli* to *E. coli* urine titers in mice and women. In the appended revised manuscript, we have addressed the comments of Reviewer 2 and included the suggested experiments that have strengthened our findings on the occurrence of VICs and the underlying mechanisms used by *E. coli* to form them.

Regarding quantification of VEC adherence and invasion in Figure 1A, 1B, and 2A, it would be useful to have a direct comparison to adherence and invasion of bladder

epithelial cells using the same MOI. These panels would also benefit from statistical analysis. Furthermore, how consistent was adherence and invasion between technical and independent replicates? Displaying the standard deviation instead of standard error of the mean may be beneficial to facilitate comparison across MOIs and between the parental strain and the type I pilus mutant.

We thank the reviewer for this recommendation. We have included results from a gentamicin protection assay comparing UPEC adherence and invasion of the 5637 bladder cell line (ATCC HTB-9) and the VK2 E6/E7 vaginal cells. We have also added a section discussing these results between the different cell types. In combination with comment 2 from Reviewer 3, we have incorporated additional experiments probing the mechanism of VEC invasion, now depicted in revised Figure 2. These new data are particularly exciting in that they demonstrate that invasion of VECs by UPEC is different from the mechanism used to invade bladder epithelial cells. As suggested by reviewer 2, we have adjusted the error bars in Fig. 1a, b to represent the standard deviation between replicates for our initial analyses for the immortalized cell line model. In agreement with reviewer 1, we “show the ideal” mean and standard error of biological replicates when comparing the mean between experimental groups as appropriate.

In Figure 2, the authors show that a type I pilus mutant (*fimA-H*) has decreased VEC adherence and invasion compared to the parental strain, but these capabilities are not completely ablated in the mutant. Are there other known UPEC receptors that are expressed on both bladder and vaginal epithelial cells that are likely to contribute to *Fim*-independent attachment?

We thank the reviewer for this comment. We now provide more evidence indicating that deletion of *fim* does not have a striking effect on vaginal colonization (revised figure 3), compared to its contribution in bladder adherence and invasion (revised figure 3g-h). These observations clearly indicate other mechanisms of bacterial adherence, which we are currently investigating. With the manuscript newly formatted as a longer Nature Communications article, we now include an expanded section, discussing potential different modes of adherence. UPEC strains can carry an abundance of different adhesive appendages; the UPEC pangenome is known to encode for 458 chaperone-usher pathway pili operons. A single UPEC strain can carry up to 16 distinct chaperone-usher pathway pili systems, we hypothesize that one or more of these systems, or potentially other adherence factors such as cellulose or curli, may contribute to vaginal cell adherence and invasion. Due to the extensive range of potentially different receptors for the interaction between UPEC bacterial cell and host vaginal cell, an extensive future study will be needed to fully understand the interaction between the cells.

Considering that type I pilus contributes to VEC invasion, is the *fimA-H* mutant still capable of colonizing the reproductive tract during urinary tract infection? Infection

studies with this mutant could further delineate the importance of vaginal colonization in seeding recurrent UTI.

We are thankful to the reviewer for this question, as it prompted us to perform experiments with the *fim* mutant that extended for longer than 24h. Specifically, we have conducted an experiment in which mice were trans-urethrally inoculated and then followed both during acute (24-hours), as well as chronic (28-days) infection stages comparing colonization of the parent and the $\Delta fimA-H$ mutant in the different genito-urinary organs. These results, now presented in revised figure 3g-h, indicate that while the *fim* mutant is significantly impaired in the bladder, it can colonize the genital organs long-term. These experiments highlight the strong stereo-chemical specificity of type 1 pili for bladder epithelial cells and strongly indicate the presence of other mechanisms that allow for persistence in the vaginal space and genital tract.

Regarding visualization of intracellular UPEC in VECs from human subjects with recurrent UTIs, these observations would be greatly strengthened by inclusion of the number of specimens that were screened and the number that had UPEC in VECs. If possible, assessment of samples from subjects without a history of recurrent UTI would also be beneficial, to determine how common *E. coli* VICs are in otherwise healthy women versus their association with recurrent UTI.

We have now added this information into the methods section. Note that in this study we sought to establish whether this previously unknown phenomenon of intracellular bacteria within vaginal cells is actually observed in humans. Indeed, in our second collected sample we identified intracellular bacteria in human vaginal samples, indicating that our observations in the lab apply to human infection. We are currently in the process of developing a method for screening for VICs in clinical vaginal swabs with higher throughput and lower false-negative rate for a long-term study looking into the prevalence of VICs among women and whether there is an association between VICs and different disease states. This ongoing study is currently outside the scope of this manuscript.

Lines 198-200: this sentence is confusing as written

We are thankful for finding the following typos in addition to suggesting the above addressed experiments. For the sake of clarification, we have revised the sentence in these lines, where it read the double negative, "UPEC had no not".

Line 203: remove the comma between "serial" and "dilution"; Line 205: "as" should be "was"

Line 208: "lyses" should be "lysis; Line 244: add "were" between "cells" and "grown"

Thank you for identifying these typographical and syntax errors. We have made all the requested changes.

Reviewer #3 (Remarks to the Author):

GENERAL:

This short report documents that UPEC can invade vaginal epithelial cells, with the authors proposing that these cells may represent an intracellular reservoir for UPEC during UTIs. The conclusions are based on studies with a vaginal epithelial cell line, a mouse transurethral infection model, and analyses of cells from patients with recurrent UTIs. The manuscript contains some compelling microscopy data, using both confocal microscopy and SEM, particularly for the in vitro cell line studies (I found the mouse model data on intracellular UPEC in Fig. 3b and 3e to be a little less convincing). The issues highlighted below should be addressed by the authors.

We are grateful to the reviewer's insightful comments that helped streamline and greatly improve the revised manuscript. The previous short manuscript was originally intended for [Redacted] as a letter and transferred to Nature Communications. We are excited to present a reformatted and extended manuscript to fit the formatting guidelines for Nature Communications. In our previous manuscript, we presented confocal, scanning (SEM) and transmission (TEM) electron microscopy and gentamicin-based protection assays to demonstrate *E. coli* VEC invasion. To further validate these observations - also as suggested by Reviewer 1 - we leveraged our highly-specific *E. coli* antibody to demonstrate intracellular *E. coli* are not labeled if the VECs are not permeabilized (Fig. 1). Additionally, we have strengthened the murine intracellular invasion, using gentamicin treatment assays of vaginal and bladder tissue to show that VICs form during acute and chronic UTI, as well as in a vaginal colonization model. Finally, we have included additional experiments, figures to address the comments of reviewer 3.

MAJOR:

1. The authors state in their opening paragraph, "Because the intestines and vagina are considered to be the primary reservoirs for UPEC - the primary cause of UTIs - infection of the urinary tract likely occurs through ascension of UPEC from these reservoirs to the urethra (reference 1)". Indeed, there is literature on the vagina as a UPEC reservoir dating back several decades, and there are several published studies demonstrating adherence of UPEC isolates to vaginal epithelial cells (VECs). Thus, the novelty in the current study appears to be restricted to the fact that UPEC can not only adhere to VECs, but also invade and survive within these cells. The findings presented certainly provide evidence of this. However, given that it is now very well-established that UPEC occupies both extracellular and intracellular niches, it is not entirely clear to this reviewer as to how novel and impactful these findings are. In my view, a better case could be made for this in the way that the authors present their findings in the manuscript. In addition,

more detailed characterization of the compartment (see below) might ensure that this work has greater influence in the field of UPEC pathogenesis.

Thank you for these comments. With the reformatting of the original manuscript to be in line with a Nature Communications article, we have addressed the reviewer's concern regarding discussing and further demonstrating the importance of reservoirs. We have included additional experiments better addressing the mechanism of invasion, the results of which are now presented in revised figure 2 and discussed extensively in the corresponding results section and discussion.

2. The authors show evidence of an intracellular niche for UPEC within VECs in Figs. 1-2. More detailed characterization of the compartment (e.g. positivity for specific endolysosomal markers such as LAMP1, PMIDs: 16968784; 19451249) and mode of entry (e.g. role of microtubules and HDAC6, PMID: 18996840), as per the existing literature on bladder epithelial cells, would be highly desirable. Functional data on specific host factors that support invasion and/or survival in VECs would greatly increase the impact of this work. Without such data, the study could be viewed as somewhat phenomenological, albeit important.

We thank reviewer 3 for this input. Guided by the suggestions of both reviewers 2 and 3, we have incorporated studies that investigate more deeply the mechanism of internalization. Specifically, we now provide data in revised Fig 2b-i that indicate a different mode of vaginal cell invasion by UPEC. Within Fig 2b-i, we present data in our pharmacological approach to determine the host signaling pathways that regulate cytoskeletal actin and microtubule re-arrangement hijacked by *E. coli* to invade VECs. Strikingly, bladder and vaginal cell invasion require active microtubule re-arrangement; though, use of histone deacetylase inhibitors do not block VEC invasion, including HDAC6 inhibitors.

3. Page 6, last 2 lines: What is the "E. coli-specific antibody" that was used? What is the epitope being recognized, and is it certain that this antibody is truly E. coli-specific? This is an important issue if claims are to be made about an intracellular UPEC reservoir in VECs of patients with recurrent UTIs. And how many patient samples are the presented data representative of (this information does not appear to be provided in Fig. 4)?

We thank reviewers 1 and 3 for raising this concern. We have included the details of this antibody within the Reporting Summary. This antibody used in this study (US Biologicals catalog number E3500-06 Lot number L8102865) is highly specific for a mixture of *E. coli* O and K antigens, the variety of O and K antigens used to generate this antibody ensure that the broad range of UPEC serotypes seen within the clinical setting can be detected. Nevertheless, we understand the reviewer's concern and have tested the antibody for cross-reactivity with *Lactobacillus crispatus*, the primary constituent of the vaginal microbiota. Figure

S1h depicts the results, demonstrating that the *E. coli* antibody is not cross-reactive with *Lactobacillus* antigens. Regarding the human sample collection; we have now added additional information about the number of patient samples included in the methods section. Note that in this study we sought to establish whether this previously unknown phenomenon of intracellular bacteria within vaginal cells is actually observed in humans. Indeed, in our second collected sample we identified intracellular bacteria in human vaginal samples, indicating that our observations in the lab apply to human infection. We are currently in the process of developing a method for screening for VICs in clinical vaginal swabs with higher throughput and lower false-negative rate for a long-term study looking into the prevalence of VICs among women and whether there is an association between VICs and different disease states. This ongoing study is currently outside the scope of this manuscript.

4. Fig. 2a shows adherence and invasion data for a FimA-H mutant. In order to be able to interpret these data appropriately, the authors should show side-by-side data with the wild type strain in the same experiments (rather than relying on comparisons with the wild type strain in different experiments – Fig. 1a). Otherwise, the data are not directly comparable.

We agree with this critique and have amended our revised manuscript as follows: To maintain organization flow of the manuscript (as also suggested by reviewer 1), we have added an additional graph in Fig. 2 to expand upon the role of type 1 pili in the early process of adherence and impact on downstream invasion. In Fig. 2, we now include a graph comparing the parental UTI89 strain with the $\Delta fimA-H$ mutant directly at a MOI of 5:1 at 120 minutes (Fig. 2a). These data indicate that type 1 pili contribute only in part to the *E. coli* adherence to VEC. This partial contribution is now further corroborated by additional experiments in which we compared the colonization of the parent and the $\Delta fimA-H$ mutant strain over time in an acute and chronic murine UTI model (Fig. 3g, h). These mouse experiments demonstrate that while deletion of *fim* impairs – as expected – colonization of the bladder, it only has no significant impact on colonization of the genital tract.

5. In Fig. 1a, intracellular UPEC loads are decreasing with increasing MOIs. This seems surprising. Was cell death monitored? Since UTI89 is a hemolysin-positive strain, is it possible that VECs are being killed during these in vitro infections? This read-out should be assessed, since it could impact on data interpretation.

Thank you for this question. In the revised manuscript we have included supplementary figure 1g that depicts cell viability information. While indeed, UTI89 is a hemolysin-positive strain, the VK2 E6/E7 cell line appears to maintain cell viability, as stated in the text, over the course of these short-term experiments focused in the immediate host-pathogen interactions. Additionally, we have included the adherence and invasion data shown in Fig. 1a and 1b shown as CFUs/mL. The cell associated CFUs are not decreasing rather the percentages are reflective of the growing population of UTI89 within the well (Fig. S1h and S1i).

6. It was not clear to this reviewer if the graphical data for in vitro infections (e.g. Fig. 1a, 1b, 1c, 2a, 2b) were combined from 3 different experiments (i.e. taking mean of each experiment) or if the data are from one experiment showing replicate wells (with similar findings to the presented data being observed in 3 different experiments). The former approach is required for appropriate statistical analysis. Can the authors clarify?

We thank the reviewer for bringing this vague description of the data to our attention. As suggested by the other reviewers as well, we have made these clarifications in that the data shown within these figures is the mean of nine independent experiments.

7. Recent studies on a mouse model to study colonization of the vagina by UPEC (PMID: 31329630) and on the impact of the vaginal microbiota on reactivation of UPEC infection in the bladder (PMID: 28358889) would seem to be relevant to this study and could be discussed in the context of the presented findings.

We agree with the relevance of these manuscripts to our work including PMID: 31329630, which was released during the review of our manuscript. We originally kept this manuscript at a limit of 1,500 words for submission as a letter to [Redacted]. Now that our manuscript has been reviewed at Nature Communications, we have expanded in the relevant areas to take advantage of the higher word count including the relevance of this PloS One article. Our data are in agreement with this recent publication that *E. coli* colonizes the vagina.

MINOR:

1. Introduction, page 3, lines 51-52: Instead of providing a single review article as a reference for the statement "Multiple studies have reported that UPEC colonizes....", it would be preferable if the authors cited multiple independent primary research papers to support this statement.

We initially cited a review for this statement to keep the reference list succinct. We have now provided a list of primary articles for referencing this statement.

2. Page 3, last line: Fig. 1c is described after Fig. 1d. The authors should either alter the text or the order of the panels in the actual figure, so that figure panels are described in order. The same issue applies to Fig. 2b (page 5, first line), which is described after Fig. 2c is presented.

We have changed the order of the panels, along with adding additional data to figure 1 and 2.

3. Fig. 1 panels and legend: What MOI was used for panel c? Please indicate this (in this panel, the level of invasion for the UTI89 control looks lower than for the data in panels

a and b?). Please also provide labels for top and bottom images in panel e. In the legend, "f" should be "e".

We thank the reviewer for bringing this to our attention. We have clarified the MOIs used within the legend. The values between the graphs are within deviations of one another; however, may appear different in value to the reviewer as the y-axis between the graphs of the two different experiments are of different length. This is now more apparent as we have shown the standard deviation in Fig. 1a, b as recommended by another reviewer.

4. In Fig. 3, arrows could be used in panel f to highlight specific examples of UPEC undergoing invagination into the VEC.

We are grateful to the reviewer for this suggestion. To make it easier to see these bacteria, we have included SEM of 100,000X magnification as well as boxed relevant bacteria.

In summary, we have addressed all of the reviewers' comments and look forward to the decision on our submitted work.

Sincerely,

Drs. Brannon and Hadjifrangiskou

REVIEWERS' COMMENTS:

Reviewer #1 (Remarks to the Author):

The authors addressed all of my previous concerns. The paper is much stronger and will be an important contribution to the field. I have only a few minor comments/suggestions.

1. The Intro sentences for both the Abstract and the Introduction, noting that most infections arise from reservoirs, still strike this reviewer as odd. The alternative is that the pathogens arise *de novo*. It may make more sense to note that most infections arise from "host-associated" reservoirs, distinguishing these from environmental reservoirs.

2. Line 149. Not clear what "heavily decorated UTI89 cells" refers to - decorated with what? Could be re-phrased for clarity.

3. Lines 126-128: Though an early study concluded that Src-family kinases are not important for UPEC invasion of bladder cells (ref 25), subsequent work (ref 46) using Src inhibitors (including PP1) indicated that FimH-mediated entry does actually require Src-family kinases. UPEC entry into vaginal epithelial cells is therefore not necessarily different from entry into bladder cells with respect to Src utilization.

4. Line 17: "...are characterized by high..."

5. Line 233: "...mediated by type 1 pili."

6. Line 239: "...invading bacteria trick host cells..." Use of word "trick" here seems too colloquial and anthropomorphic.

7. Line 251-252 is redundant with preceding sentence.

8. Lines 262-263. Not clear how "bacteriuria within these mice is not unlike clinical cases of ASB...". Could clarify this by reminding reader that low urine titers were often present in mice with high titers in vaginal tract. However, is it clear that these mice are asymptomatic for UTI, even though urine titers are low? Seems more work would be needed to substantiate this statement. Probably better to just clarify and qualify the statement.

Reviewer #2 (Remarks to the Author):

The authors were very responsive to all reviewer comments, and the revised manuscript includes new data that strengthens the conclusions. This is an excellent and compelling body of work.

Very minor comments:

line 28: there should be a comma after "the gut"

line 233: remove "of" before "type I pili"

Reviewer #3 (Remarks to the Author):

GENERAL:

The authors have gone to considerable lengths to address the comments of all three reviewers, and I feel that this revised version is much improved in placing the current study in the context of the existing literature and in addressing some experimental limitations that were apparent in the

original submission. I do still have some relatively minor comments that I feel should be addressed by the authors.

SPECIFIC COMMENTS:

1. The broad rationale for experiments with various inhibitors to assess mechanisms of VEC invasion (Figure 2d-i) is generally sound, but I have some concerns about approach, controls or data interpretation. For example, differences are claimed between effects of genistein and wortmannin in invasion, yet no data are presented on actual target inhibition in these cells and genistein does seem to have some effect on cell viability by comparison to wortmannin – see Fig. S1g, which is on a log scale. Thus, could reduced invasion for genistein-treated cells reflect greater cell death or general state of the cells? Moreover, in comparing effects of genistein and wortmannin the authors state “These results suggest that other host tyrosine kinases, besides PI3K are essential to VEC invasion.” While genistein is a tyrosine kinase inhibitor, PI3K is a lipid kinase and its major downstream target is Akt (a serine/threonine kinase). Thus, the wording of this sentence needs to be modified. Also in relation to the results text describing Figure 2, trichostatin A is a pan-HDAC inhibitor, rather than being an HDAC6/HDAC10-selective inhibitor - so this text needs to be modified (tubastatin A would have been a more appropriate choice as an HDAC6-selective inhibitor). Demonstration of target inhibition for those compounds not having any effect on invasion would have been ideal to ensure that the presented data are being appropriately interpreted (e.g. showing that TSA caused tubulin hyperacetylation - read-out of HDAC6 inhibition - yet had no effect on invasion). These are important issues, since the authors make claims about mechanistic differences between VECs and bladder epithelial cells, yet they don't actually assess the effects of the various inhibitors on bladder epithelial cells. It is appreciated that additional experiments may not be feasible given current circumstances, but at the very least the authors should carefully review/modify the text in this section for accuracy (e.g. for compound specificity) and to remove any over-interpretation of findings - toning down claims about mechanistic differences between VECs and bladder epithelial cells in some places may be appropriate in the absence of direct comparisons between these cell types for effects of the various inhibitors.

2. A previous minor point requiring correction was that Fig. 1c was described after Fig. 1d in the original manuscript - this still remains the case in the revised version, despite the author's assertion that this has been corrected. Figure panels should be presented in the order that they are cited (currently Fig. 1d is mentioned before Fig. 1c). The same issue applies for Supplementary Fig. 1j, which is presented in the text as the first Supplementary Figure panel. It would seem appropriate for the authors to check all in-text citations of figures and revise either text or figure panels, so that each figure panel is introduced in order (i.e. don't introduce Fig. 1d before Fig. 1c or Supplementary Fig. 1j before Supplementary Fig. 1a-i). Please also check that figures have been cited correctly – for example, line 316 (pg 16) cites Supplementary Fig. 2S, but I think it should actually refer to Supplementary Fig. S1g?

REVIEWERS' COMMENTS:

Reviewer #1 (Remarks to the Author):

The authors addressed all of my previous concerns. The paper is much stronger and will be an important contribution to the field. I have only a few minor comments/suggestions.

1. The Intro sentences for both the Abstract and the Introduction, noting that most infections arise from reservoirs, still strike this reviewer as odd. The alternative is that the pathogens arise de novo. It may make more sense to note that most infections arise from "host-associated" reservoirs, distinguishing these from environmental reservoirs.

We have addressed the reviewer's concern and have provided clarity regarding the type of reservoir. Where the first sentence in the Abstract (line 2) read, "Pathogen reservoirs account..." now reads as, "Host-associated reservoirs account..." We kept the first sentence of the Introduction (lines 15-16) the same as it already specifies, "bacterial reservoirs, particularly those within the host".

2. Line 149. Not clear what "heavily decorated UTI89 cells" refers to - decorated with what? Could be re-phrased for clarity.

We have provided clarity to this sentence to specify pili. The sentence now reads as, "UTI89 cells that appear to be heavily decorated with pili".

3. Lines 126-128: Though an early study concluded that Src-family kinases are not important for UPEC invasion of bladder cells (ref 25), subsequent work (ref 46) using Src inhibitors (including PP1) indicated that FimH-mediated entry does actually require Src-family kinases. UPEC entry into vaginal epithelial cells is therefore not necessarily different from entry into bladder cells with respect to Src utilization.

We are thankful for the reviewer for bringing this point to the discussion. We have referenced the mixed results of the two previous studies to which the authors noted the reasons for the different results between the two studies are unknown (ref 46). We have softened the phrasing in this sentence (lines 126-128) concerning the difference in Src-family kinase roles in bladder and vaginal cell invasion concerning these previous studies.

4. Line 17: "...are characterized by high..."

We have corrected this typo and replaced "is" with "are"

5. Line 233: "...mediated by type 1 pili."

We have corrected this type and deleted the word "of"

6. Line 239: "...invading bacteria trick host cells..." Use of word "trick" here seems too colloquial and anthropomorphic.

We have rewritten this sentence to remove the word "trick".

7. Line 251-252 is redundant with preceding sentence.

We thank the reviewer for catching this redundancy and have merged the two sentences together to be concise.

8. Lines 262-263. Not clear how "bacteriuria within these mice is not unlike clinical cases of ASB...". Could clarify this by reminding reader that low urine titers were often present in mice with high titers in vaginal tract. However, is it clear that these mice are asymptomatic for UTI, even though urine titers are low? Seems more work would be needed to substantiate this statement. Probably better to just clarify and qualify the statement.

We are thankful for the reviewer for pointing out the vague phrase. We have clarified our reference to the mouse titers in this sentence reflecting the low urine titers in ASB cases. Additionally, we have provided a reference for the Infectious Diseases Society of America definition of ASB.

Reviewer #2 (Remarks to the Author):

The authors were very responsive to all reviewer comments, and the revised manuscript includes new data that strengthens the conclusions. This is an excellent and compelling body of work.

Very minor comments:

We thank the reviewer for catching these typos and have made the corrections.

line 28: there should be a comma after "the gut"

line 233: remove "of" before "type I pili"

Reviewer #3 (Remarks to the Author):

GENERAL:

The authors have gone to considerable lengths to address the comments of all three reviewers, and I feel that this revised version is much improved in placing the current study in the context of the existing literature and in addressing some experimental limitations that were apparent in the original submission. I do still have some relatively minor comments that I feel should be addressed by the authors.

SPECIFIC COMMENTS:

1. The broad rationale for experiments with various inhibitors to assess mechanisms of VEC invasion (Figure 2d-i) is generally sound, but I have some concerns about approach, controls or data interpretation. For example, differences are claimed between effects of genistein and wortmannin in invasion, yet no data are presented on actual target inhibition in these cells and genistein does seem to have some effect on cell viability by comparison to wortmannin – see Fig. S1g, which is on a log scale. Thus, could reduced invasion for genistein-treated cells reflect greater cell death or general state of the cells? Moreover, in comparing effects of genistein and wortmannin the authors state “These results suggest that other host tyrosine kinases, besides PI3K are essential to VEC invasion.” While genistein is a tyrosine kinase inhibitor, PI3K is a lipid kinase and its major downstream target is Akt (a serine/threonine kinase). Thus, the wording of this sentence needs to be modified. Also in relation to the results text describing Figure 2, trichostatin A is a pan-HDAC inhibitor, rather than being an HDAC6/HDAC10-selective inhibitor - so this text needs to be modified (tubastatin A would have been a more appropriate choice as an HDAC6-selective inhibitor). Demonstration of target inhibition for those compounds not having any effect on invasion would have been ideal to ensure that the presented data are being appropriately interpreted (e.g. showing that TSA caused tubulin hyperacetylation - read-out of HDAC6 inhibition - yet had no effect on invasion). These are important issues, since the authors make claims about mechanistic differences between VECs and bladder epithelial cells, yet they don't actually assess the effects of the various inhibitors on bladder epithelial cells. It is appreciated that additional experiments may not be feasible given current circumstances, but at the very least the authors should carefully review/modify the text in this section for accuracy (e.g. for compound specificity) and to remove any over-interpretation of findings - toning down claims about mechanistic differences between

VECs and bladder epithelial cells in some places may be appropriate in the absence of direct comparisons between these cell types for effects of the various inhibitors.

We are thankful to the reviewer's positive response to the inclusion of these experiments and attention to detail. If these differences in invasion were a result of cell death or state, we would have observed a reduction in bacterial adherence. In Fig. S1b,c, we show that there is no statistically significant difference in *E. coli* adherence. Additionally, as noted in Fig. S1g there is no significant difference between genistein and wortmannin and controls as determined by non-parametric Kruskal-Wallis test. We are thankful to the reviewer for catching the error in the sentence, "These results suggest that other host tyrosine kinases, besides PI3K are essential to VEC invasion.". The reviewer is correct in that PI3K is a lipid kinase, which we refer to as such on line 120. As the reviewer has suggested, we have modified this sentence by removing the word "tyrosine" from line 125. We appreciate the reviewer's comment regarding are simplification of trichostatin A, though it is not a broad pan-HDAC inhibitor. We've included reference regarding trichostatin inhibition of HDACs (DOI 10.1242/jcs.046813). We have also modified the section for accuracy as suggested by the reviewer. We have done so to more clearly highlight that between the combination of experiments using butyrate (does not inhibit HDAC6 and HDAC10) and trichostatin A (does inhibit HDAC6 and HDAC10) we were able to rule out a broader range of HDACs being involved in VEC invasion. As suggested by the reviewer, we have carefully toned down the findings of the experiments in this paragraph.

2. A previous minor point requiring correction was that Fig. 1c was described after Fig. 1d in the original manuscript - this still remains the case in the revised version, despite the author's assertion that this has been corrected. Figure panels should be presented in the order that they are cited (currently Fig. 1d is mentioned before Fig. 1c). The same issue applies for Supplementary Fig. 1j, which is presented in the text as the first Supplementary Figure panel. It would seem appropriate for the authors to check all in-text citations of figures and revise either text or figure panels, so that each figure panel is introduced in order (i.e. don't introduce Fig. 1d before Fig. 1c or Supplementary Fig. 1j before Supplementary Fig. 1a-i). Please also check that figures have been cited correctly – for example, line 316 (pg 16) cites Supplementary Fig. 2S, but I think it should actually refer to Supplementary Fig. S1g?

We appreciate the reviewer for catching the difference in Figure number order. We have swapped the placement of the order of Fig. 1c and 1d to fix this. We have also updated the ordering of Supplementary Fig. 1 to match the order in the manuscript.